# Integrated healthy lifestyle even in late-life mitigates cognitive decline risk across varied genetic susceptibility

Jun Wang[1,2], Chen Chen [1,2], Jinhui Zhou [1,3], Zinan Xu [1,4], Lanjing Xu[1,5], Xinwei Li[1,6], Zhuchun Zhong[1,7], Yuebin Lv [1,2,8] ✉ & Xiaoming Shi [1,2,8] ✉

It remains unclear whether the benefits of adhering to a healthy lifestyle outweigh the effects of high genetic risk on cognitive decline. We examined the association of combined lifestyle factors and genetic risk with changes in cognitive function and six specific dimensions of cognition among older adults from the Chinese Longitudinal Healthy Longevity Survey (1998–2018, $n = 18{,}811$, a subset of 6301 participants with genetic information). Compared to participants with an unfavorable lifestyle, those with a favorable lifestyle showed a 46.81% slower rate of cognitive decline, with similar results across most cognitive domains. High genetic risk was associated with a 12.5% faster rate of cognitive decline. Individuals with a high genetic risk and a favorable lifestyle have slower cognitive decline than those with a low genetic risk and an unfavorable lifestyle. These data suggest that the benefits of a favorable lifestyle outweigh genetic factors, and therefore that adhering to a favorable lifestyle may offset the genetic risk for accelerated cognitive decline.

The global number of people with dementia is projected to reach 152.8 million by 2050, nearly triple the 2019 estimate of 57.4 million, posing a significant public health threat worldwide[1]. Notably, China has the largest population of patients with dementia in the world, bringing an enormous public health burden[2]. Mild cognitive impairment is considered as an early stage of dementia[3], with up to 15% of individuals with mild cognitive impairment progressing to dementia within two years[4]. Preventing or delaying cognitive decline and dementia could significantly improve quality of life and prolong functional independence in late life[3].

Currently, there are no effective drugs to prevent, delay, or cure dementia. Therefore, identifying measures to reduce the risk of its onset is crucial to better disease management, benefiting both individuals and society[3]. Accumulating evidence demonstrated that adherence to a healthy lifestyle, as a key modifiable risk factor, including a high-quality diet intake, active physical activity, never smoking, and limited drinking, was associated with better cognitive status and may prevent cognitive impairment[5–9]. The 2020 recommendations of The Lancet Commission suggest that addressing modifiable risk factors might prevent or delay up to 40% of dementia cases[10], a view also supported by the World Health Organization[11]. Apart from modifiable risk factors, genetic factors have also been linked to cognitive impairment[12–14]. Previous studies have found that adherence to a healthy lifestyle was associated with slower cognitive decline, regardless of the presence of the apolipoprotein E *(APOE) ε4* allele[15,16]. Moreover, a large and still-growing number of genetic

[1]China CDC Key Laboratory of Environment and Population Health, National Institute of Environmental Health, Chinese Center for Disease Control and Prevention, Beijing, China. [2]National Key Laboratory of Intelligent Tracking and Forecasting for Infectious Diseases, National Institute of Environmental Health, Chinese Center for Disease Control and Prevention, Beijing, China. [3]National Cancer Center/National Clinical Research Center for Cancer/Cancer Hospital, Chinese Academy of Medical Sciences and Peking Union Medical College, Beijing, China. [4]Department of Epidemiology, School of Public Health, Southern Medical University, Guangzhou, Guangdong, China. [5]Department of Big Data in Health Science, School of Public Health, Zhejiang University, Hangzhou, Zhejiang, China. [6]Department of Epidemiology and Biostatistics, School of Public Health, Jilin University, Changchun, Jilin, China. [7]Institute of Environmental Medicine, Zhejiang University School of Medicine, Hangzhou, Zhejiang, China. [8]These authors jointly supervised this work: Yuebin Lv, Xiaoming Shi. ✉e-mail: lvyuebin@nieh.chinacdc.cn; shixm@chinacdc.cn

variants associated with cognitive function has been identified beyond *APOE ε4*[14]. Genetic risk score, which combines information from multiple genetic markers, has been developed to better predict the risk of cognitive impairment[13]. Evidence from the Rotterdam Study reported that the benefits of adhering to a favorable lifestyle on lowering dementia risk were more pronounced among people with low/intermediate genetic risk than those at high risk[17]. However, it is unknown whether the impact of lifestyle factors on cognitive decline differs by genetic risk. Additionally, given the ancestry difference in the progression of cognitive decline and lifestyle habits, more research is required to understand the effect of lifestyle factors and genetic risk on the patterns of cognitive change among older Chinese adults.

Therefore, the aim of the present study is to assess the role of combined lifestyle factors and genetic risk on the decline rate of overall cognitive function and specific cognitive domains among Chinese older adults in a population-based cohort study, the Chinese Longitudinal Healthy Longevity Survey (CLHLS). In this work, we show that a favorable lifestyle has a stronger impact than genetic factors in slowing cognitive decline, supporting lifestyle interventions for older adults at risk for cognitive decline.

## Results

Baseline characteristics of study participants are shown in Table 1. Among 18,811 participants (mean age 82.97 years, 52.28% women), 10,515 (55.90%) were of unfavorable lifestyle, 5885 (31.28%) of intermediate lifestyle, and 2411 (12.82%) of favorable lifestyle. The mean (SD) value of baseline cognitive score was 26.73 (3.24). The present study involved 6301 participants with genetic information, and the polygenic risk score was approximately normally distributed (Supplementary Fig. 1). Participants were divided equally into low-genetic risk and high-genetic risk groups. Compared to participants with unfavorable lifestyles, those with favorable lifestyles were more likely to be younger, female, married, urban residents, and to have higher educational attainment (Supplementary Data 1). Most baseline characteristics by status of genetic risk were similar between the two groups (Supplementary Data 1). As compared with participants with unfavorable lifestyles, those with favorable lifestyles had a greater adjusted mean of the cognitive score at baseline (Supplementary Fig. 2), that is, 0.54 higher for the overall cognitive score, 0.27 higher for the orientation score, 0.33 higher for the attention and calculation score, 0.42 higher for the visual construction score, 0.27 higher for the language score, 0.25 higher for the naming score, and 0.31 higher for the recall score, respectively. Similar findings were observed, where an increasing number of favorable lifestyle factors was associated with a better cognitive score at baseline within both genetic risk groups.

### Individual association of healthy lifestyle and genetic risk with cognitive decline

Table 2 shows the rate of decline in cognitive score for participants with different lifestyle profiles. The rate of cognitive decline for participants with intermediate lifestyles was 15.96% slower (−0.316 standard deviation unit [SDU] per year [95% CI, −0.333 to −0.299 SDU per year]; difference, 0.060 SDU per year [95% CI, 0.039–0.082 SDU per year]), and the rate of cognitive decline for participants with favorable lifestyles was 46.81% slower (−0.200 SDU per year [95% CI, −0.226 to −0.174 SDU per year]; difference, 0.176 SDU per year [95%CI, 0.147–0.205 SDU per year]), compared with those with unfavorable lifestyles (−0.376 SDU per year [95% CI, −0.390 to −0.363 SDU per year]). Similar results were obtained in the analyses for each specific dimension of cognitive function. Participants with favorable lifestyles showed a 50.92% slower decline in orientation score, 40.3% slower decline in attention and calculation score, 23.08% slower decline in visual construction score, 51.24% slower decline in language score, 49.34% slower decline in naming score, and 43.05% slower decline in recall score, respectively. Participants with unfavorable lifestyles

had the steepest declines in cognitive function and the lowest cognitive scores over time compared with the other two lifestyle groups (Supplementary Fig. 3). The potential cognitive health benefits of adhering to favorable lifestyle were greater for females than for males (*P* interaction < 0.001, Supplementary Fig. 4).

The cognitive decline occurred slower in participants with low genetic risk than those with high genetic risk (−0.161 SDU per year [95% CI, −0.176 to −0.146 SDU per year] vs. −0.184 SDU per year [95% CI, −0.199 to −0.169 SDU per year]; difference, 0.023 SDU per year [95% CI, 0.003–0.044 SDU per year], Supplementary Table 1).

### Association of healthy lifestyle with cognitive decline stratified by genetic risk

Stratified analyses showed that participants with favorable lifestyles had significantly slower rates of cognitive decline compared with those with unfavorable lifestyles for both low and high-genetic risk groups (Fig. 1). For participants with low genetic risk, the rate of cognitive decline for those with favorable lifestyles was 42.7% slower (difference, 0.079 SDU per year [95%CI, 0.040–0.118 SDU per year]) compared with those with unfavorable lifestyles (Supplementary Data 2). Among participants with high genetic risk, those with favorable lifestyles had a 32.5% slower cognitive decline (difference, 0.065 SDU per year [95% CI, 0.023–0.107 SDU per year]) compared with those with unfavorable lifestyles. However, no significant interaction effect between genetic risk and lifestyle on cognitive decline was observed (*P* interaction = 0.29). The benefits of a healthy lifestyle on change of orientation score and language score in both genetic risk groups remained statistically significant after applying Bonferroni multiple correction (*P* < 0.008).

### Joint association of healthy lifestyle and genetic risk with cognitive decline

Compared with participants with high genetic risk and unfavorable lifestyle (Supplementary Data 3), those with low genetic risk and favorable lifestyle had 44.9% slower rate of decline for cognitive score (difference, 0.088 SDU per year [95% CI, 0.047–0.129 SDU per year]), 45.6% slower rate of decline for orientation score (difference, 0.088 SDU per year [95% CI, 0.042–0.135 SDU per year]), 35% slower rate of decline for attention and calculation score (difference, 0.028 SDU per year [95% CI, 0.008–0.048 SDU per year]), 56.77% slower rate of decline for language score (difference, 0.088 SDU per year [95% CI, 0.051–0.125 SDU per year]), 44.62% slower rate of decline for naming score (difference, 0.029 SDU per year [95% CI, 0.005–0.052 SDU per year]), and 33.33% slower rate of decline for recall score (difference, 0.026 SDU per year [95%CI, 0.003–0.050 SDU per year]), respectively. Figure 2 shows a steeper decline in most cognitive functions, with the exception of the visual construction, among participants with high genetic risk and unfavorable lifestyle, and a more obvious change of slopes between groups with favorable and unfavorable lifestyles than that with different genetic risks.

### Association of healthy lifestyle and genetic risk with cognitive impairment risk

Compared with unfavorable lifestyles, adhering to favorable lifestyles was associated with a lower risk of cognitive impairment, with an adjusted hazard ratio (HR) and 95% CI of 0.69 (0.62–0.76) (Supplementary Table 2). Individuals at high genetic risk had a 10% higher risk of cognitive impairment than those with low genetic risk (HR = 1.10, 95% CI, 1.00–1.21). Compared with participants with low genetic risk and favorable lifestyle (Supplementary Table 3), those with high genetic risk and unfavorable lifestyle experienced the highest risk of incident cognitive impairment (HR = 1.41, 95% CI, 1.11–1.78). Stratified analysis showed that both favorable and intermediate lifestyles were related to a lower risk of cognitive impairment among individuals at low genetic risk, while the protective associations were only evident in favorable profiles among those with high genetic risk (Supplementary

## Table 1 | Baseline characteristics of the included participants

| Characteristic | All participants (N = 18,811) | Participants with genetic information (N = 6301) |
|---|---|---|
| Age, mean ± SD, years | 82.97 ± 10.65 | 79.13 ± 10.44 |
| Sex, N (%) | | |
| Male | 8976 (47.72) | 3159 (50.13) |
| Female | 9835 (52.28) | 3142 (49.87) |
| Area of residence, N (%) | | |
| Urban | 7874 (41.86) | 2263 (35.91) |
| Rural | 10,937 (58.14) | 4038 (64.09) |
| Educational attainment, N (%) | | |
| <1 year | 10,612 (56.41) | 3331 (52.86) |
| 1–6 years | 6043 (32.12) | 2153 (34.17) |
| > 6 years | 2156 (11.46) | 817 (12.97) |
| Source of income, N (%) | | |
| Dependent | 12,815 (68.13) | 3944 (62.59) |
| Independent | 5996 (31.87) | 2357 (37.41) |
| Marital status, N (%) | | |
| Not in marriage | 11,166 (59.36) | 3086 (48.98) |
| In marriage | 7645 (40.64) | 3215 (51.02) |
| Occupation, N (%) | | |
| Agriculture/forestry/husbandry/fishery | 11,488 (61.07) | 4220 (66.97) |
| Commercial, service, or industrial worker/self-employer | 3208 (17.05) | 998 (15.84) |
| Professional/governmental/managerial personnel | 1809 (9.62) | 606 (9.62) |
| Houseworker/never worked/other | 2306 (12.26) | 477 (7.57) |
| Healthy lifestyle factors, N (%) | | |
| Unfavorable | 10,515 (55.90) | 3111 (49.37) |
| Intermediate | 5885 (31.28) | 2146 (34.06) |
| Favorable | 2411 (12.82) | 1044 (16.57) |
| Baseline cognitive score[a], mean ± SD | 26.73 ± 3.24 | 27.21 ± 3.01 |
| Orientation score | 4.84 ± 0.51 | 4.89 ± 0.41 |
| Attention and calculation score | 4.32 ± 1.38 | 4.44 ± 1.26 |
| Visual construction score | 0.35 ± 0.48 | 0.39 ± 0.49 |
| Language score | 5.70 ± 0.78 | 5.77 ± 0.67 |
| Naming score | 6.35 ± 1.29 | 6.43 ± 1.18 |
| Recall score | 5.18 ± 1.29 | 5.30 ± 1.23 |

Mean (SD) for continuous variables and number (percentage) for dichotomous variables.
N sample size, SD standard deviation.
[a]The raw value of the cognitive score.

Table 4, P interaction = 0.016). Additional accounting for competing risk of death slightly attenuated the association (Supplementary Tables 2–4).

## Sensitivity analyses

The strength and magnitude of association between the healthy lifestyle with cognitive decline and its respective domains among overall participants and different genetic risk groups remained largely unchanged in a series of sensitivity analyses (Supplementary Data 4, 5, Supplementary Table 5, Supplementary Fig. 5). When a quadratic term for the time was included in the multivariate linear mixed-effects models, the fitted curves were not perfectly straight but displayed slightly curved (Fig. 3).

## Discussion

Using population-based longitudinal cohort study of older adults with repeated measurements, we found that cognitive function decreased over time. However, adherence to a favorable lifestyle was associated with a slower decline in cognitive function and specific aspects of cognition, such as orientation and language ability. Moreover, the beneficial effects of favorable lifestyle on delaying cognitive decline were observed within both genetic risk categories. Our findings suggest that a healthy lifestyle might outweigh genetic factors on cognitive decline, highlighting the particular importance of adherence to a healthy lifestyle in the maintenance of long-term cognitive performance.

Previous research found that adherence to a healthy lifestyle was associated with lower risks of developing cognitive impairment and dementia[18–20]. Likewise, several studies have also shown a healthy lifestyle was linked to slower cognitive decline over time[18,21]. However, studies investigating associations between individual lifestyle profiles and changes in cognitive function have yielded inconsistent results[22–26]. A possible explanation for these discrepancies is the clustering of healthy behaviors[27]. For example, participants with a healthy diet intake are more likely to engage in other healthy lifestyle behaviors (e.g., abstaining from smoking and active physical activity), making it difficult to tease apart the independent outcomes of diet intake, in particular when cognitive performance is influenced by multiple lifestyle behaviors[24].

As the *APOE ε4* allele is considered to be one of the major genetic risk factors for cognitive decline and dementia[28], most of the research has been focused on examining the effect of a healthy lifestyle and *APOE ε4* on the annual rate of change in cognition. However, the results were inconsistent[15,16,29,30]. For example, the Chicago Health and Aging Project revealed that adherence to a healthy lifestyle was related to a slower rate of cognitive decline in both *APOE ε4* carriers ($\beta = 0.029$, 95% CI, 0.013–0.045) and noncarriers ($\beta = 0.013$, 95% CI, 0.005–0.022)[16], with similar findings observed across different ancestry groups in the following study[15]. Recently, Jia et al. evaluating the cognitive trajectories among cognitively normal Chinese older adults, reported that the overall cognition state remained stable, whereas memory declined continuously over time. A healthy lifestyle was associated with slower memory decline, even in the presence of the *APOE ε4* allele[30]. These inconsistencies across studies might be partly explained by variations in study design, including differences in the ancestry background of the study population, sample size of the study, and participants' characteristics.

Cognitive decline, like most other complex traits, is influenced by a number of variants[15–17,29,31,32]. Studies investigating the associations between healthy lifestyle and weighted genetic risk score on change in cognitive score and specific cognitive domains are limited. In this study, we used data from a population-based sample without cognitive impairment at baseline to compute a weighted genetic risk score based on multiple genetic variants, a preferred method in gene–environment interactions analyses. We reported that individuals with high genetic risk and favorable lifestyle experienced slower cognitive decline compared with those at low genetic risk and unfavorable lifestyle, and similar effects were observed for some of the cognitive subdomains, such as orientation and language ability. These findings suggest that adherence to a favorable lifestyle might generate a greater beneficial effect on cognitive decline than those of genetic factors. The modest effects of genetic factors on cognitive decline may be due in part to epigenetic changes that altered the transcription of risk genes for Alzheimer's disease[14,33]. From a public health perspective, it might be feasible to encourage populations to improve their lifestyles for the primary prevention of cognitive decline, particularly among individuals with high genetic risk. Moreover, our study comprised predominantly oldest-old, and the benefits of adhering to a healthy lifestyle for better cognitive function persist in the oldest-old. These

**Table 2 | Association of healthy lifestyle with the rate of cognitive decline among overall participants**

| Lifestyle | Standard deviation units | | | Difference (%) |
|---|---|---|---|---|
| | Estimate (95% CI) | Difference (95% CI) | *P* value | |
| **Cognitive score** | | | | |
| Unfavorable | −0.376 (−0.390, −0.363) | Reference | | Reference |
| Intermediate | −0.316 (−0.333, −0.299) | 0.060 (0.039, 0.082) | <0.001 | 15.96 |
| Favorable | −0.200 (−0.226, −0.174) | 0.176 (0.147, 0.205) | <0.001 | 46.81 |
| **Orientation score** | | | | |
| Unfavorable | −0.379 (−0.395, −0.363) | Reference | | Reference |
| Intermediate | −0.316 (−0.337, −0.295) | 0.063 (0.036, 0.089) | <0.001 | 16.62 |
| Favorable | −0.186 (−0.218, −0.155) | 0.193 (0.157, 0.228) | <0.001 | 50.92 |
| **Attention and calculation score** | | | | |
| Unfavorable | −0.134 (−0.140, −0.127) | Reference | | Reference |
| Intermediate | −0.112 (−0.120, −0.104) | 0.022 (0.012, 0.032) | <0.001 | 16.42 |
| Favorable | −0.080 (−0.091, −0.068) | 0.054 (0.041, 0.067) | <0.001 | 40.3 |
| **Visual construction score** | | | | |
| Unfavorable | −0.013 (−0.017, −0.009) | Reference | | Reference |
| Intermediate | −0.011 (−0.015, −0.006) | 0.002 (−0.004, 0.009) | 0.425 | 15.38 |
| Favorable | −0.010 (−0.017, −0.003) | 0.003 (−0.005, 0.011) | 0.426 | 23.08 |
| **Language score** | | | | |
| Unfavorable | −0.322 (−0.335, −0.309) | Reference | | Reference |
| Intermediate | −0.247 (−0.263, −0.230) | 0.075 (0.054, 0.095) | <0.001 | 23.29 |
| Favorable | −0.156 (−0.181, −0.132) | 0.165 (0.138, 0.193) | <0.001 | 51.24 |
| **Naming score** | | | | |
| Unfavorable | −0.152 (−0.160, −0.144) | Reference | | Reference |
| Intermediate | −0.120 (−0.131, −0.110) | 0.031 (0.019, 0.044) | <0.001 | 20.39 |
| Favorable | −0.077 (−0.092, −0.062) | 0.075 (0.058, 0.092) | <0.001 | 49.34 |
| **Recall score** | | | | |
| Unfavorable | −0.151 (−0.159, −0.144) | Reference | | Reference |
| Intermediate | −0.127 (−0.137, −0.118) | 0.024 (0.012, 0.035) | <0.001 | 15.89 |
| Favorable | −0.086 (−0.100, −0.072) | 0.065 (0.049, 0.080) | <0.001 | 43.05 |

Linear mixed-effects models were used with adjustment for age, sex, entry time, educational attainment, area of residence, current marital status, occupation, source of income, and baseline cognitive score. For the analysis of cognitive dimensions, models were additionally adjusted for the baseline dimensions of cognitive score as appropriate instead of baseline cognitive score. Two-sided $P < 0.05$ was considered statistically significant, except for separate analysis for individual domains of cognition in which the Bonferroni correction was applied to account for multiple testing ($P < 0.008$ considered significant [=0.05/6]).
*CI* confidence interval.

findings align with the latest study showing that a healthy lifestyle was associated with better cognitive function proximate to death, with an average death age of 90 years, independently of common neuropathologies of dementia[34]. A recent systematic review found that individual lifestyle factors could reduce the risk of developing cognitive impairment and dementia even among the oldest-old, while the evidence about the effect of combined lifestyle on the rate of cognitive decline in oldest-old is insufficient[9].

The Finnish Geriatric Intervention Study to Prevent Cognitive Impairment and Disability (FINGER) has recently tested the effect of multidomain intervention on cognitive performance among 1259 older adults, indicating that improved lifestyles were linked to cognitive improvement[35]. However, evidence from a systematic review of randomized controlled trials on the relationship of lifestyle intervention with cognition change was conflicting, which may be partially due to the substantial heterogeneity in intervention content, delivery and duration, target populations, and outcome measures[36].

Strengths of this study include the availability of follow-up data to estimate the annual changes in cognitive function and its respective domains over time and the exclusion of individuals with cognitive impairment or dementia at baseline to reduce the possibility of reverse causality. The limitations of this study should also be noted. First, information on healthy lifestyle factors and cognitive function

measures was self-reported; thus, measurement errors are inevitable. However, these data have been well validated in previous studies[7,37–39]. Second, the dietary information collected was insufficient to estimate total energy intake and overall healthy dietary behaviors, limiting to further adjusting for energy intake and exactly providing a more comprehensive picture of dietary practices. Future studies with expanded information on dietary intake would be needed. Third, participants were categorized according to baseline lifestyle score, which could not capture the long-term cumulative effects of lifestyle factors and might underestimate the association[40]. However, the impact of lifestyle trajectory groups on cognitive decline was evaluated, and the results were in line with the main analysis. Fourth, although our analyses were simultaneously adjusted for multiple factors to reduce the influence of residual confounding, some unmeasured or unknown factors may have accounted for the association we found. Fifth, given the limited number of items to assess the visual construction and common occurrence of missing values in this domain[41], a nonsignificant association between a healthy lifestyle and genetic risk on visual construction should be interpreted with caution. Further studies using multiple indicators to evaluate visuospatial ability are needed. Additionally, being restricted to participants who completed all Mini-Mental State Examination (MMSE) items might lead to an overestimation of the association between healthy lifestyle and

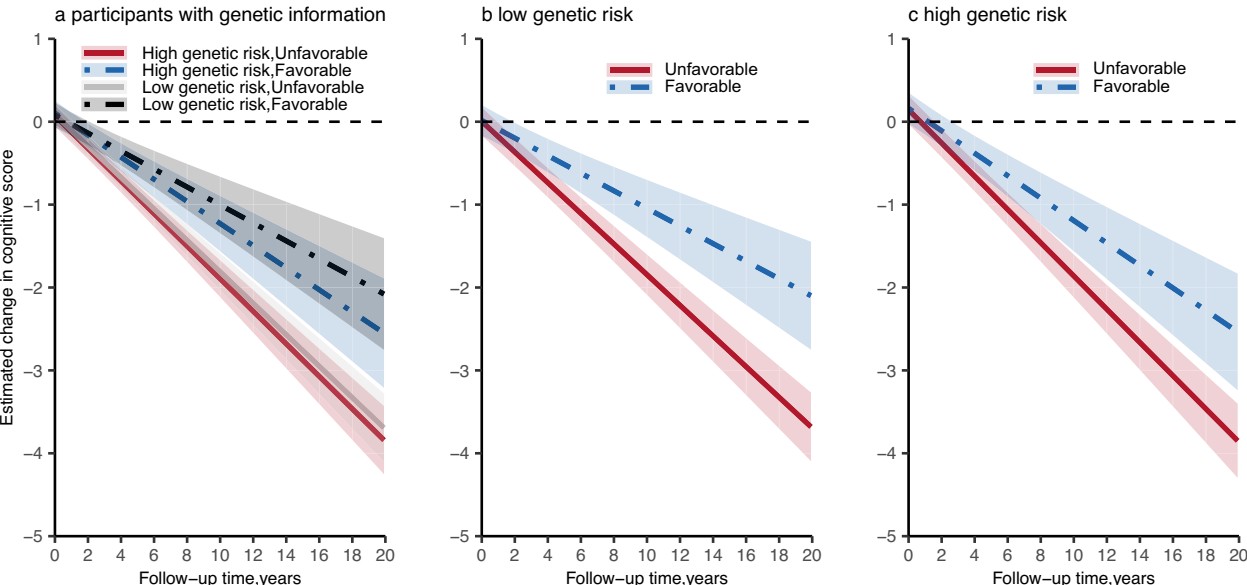

**Fig. 1 | Rate of change in cognitive function according to genetic risk and lifestyle profiles.** Linear mixed-effects models were used with adjustment for age, sex, entry time, educational attainment, area of residence, current marital status, occupation, source of income, and baseline cognitive score. **a** Joint effect of lifestyle and genetic risk on cognitive decline, **b** among participants with low genetic risk, the association of lifestyle with cognitive decline, **c** among participants with high genetic risk, the association of lifestyle with cognitive decline. In **a** solid red line represents point estimates of high genetic risk and unfavorable lifestyle group, dashed blue line represents point estimates of high genetic risk and favorable lifestyle group, a solid gray line represents point estimates of low genetic risk and unfavorable lifestyle group, dashed black line represents point estimates of low genetic risk and favorable lifestyle group, and shaded areas show 95% confidence interval (CI). In **b** and **c**, the solid red line represents point estimates of unfavorable lifestyle groups, the dashed blue line represents point estimates of favorable lifestyle groups, and shaded areas show 95% CI. Source data are provided as a Source Data file.

cognitive decline[41]. However, the results of the sensitivity analysis, including only participants who completed all MMSE items, were consistent with the main analysis. Sixth, our genetic risk score did not include all the variants associated with cognitive decline, which might attenuate the true effect. Future studies should construct genetic risk scores of cognitive decline by comprehensive coverage of established cognitive decline-related genetic variants to improve the accuracy of risk classification. Additional studies with larger sample sizes are warranted to verify our findings and examine the impact of genetic risk factors on cognitive decline at the extreme end of the polygenic risk score distribution. Seventh, individuals with risk alleles and an unhealthy lifestyle have an increased risk of premature mortality and may have died before participating in our study or died prior to follow-up investigation, which might introduce potential survival bias. Moreover, MMSE was the only measure of cognitive function in our study, and we cannot exclude the possibility that including other neurocognitive tests might yield different results. Eighth, although the possibility of a non-linear association between lifestyle and cognitive decline has been observed, the fitted curve was highly close to the straight line, which needs to be validated in further studies. Another significant limitation is that the study sample included primarily individuals with low educational levels, thus limiting the generalizability of our findings to those with higher levels of education. Finally, our results were restricted to the Chinese older adults, and it remains to be examined whether our results could be generalized to other ethnic populations.

In summary, our study suggests even in late life, adhering to a favorable lifestyle was associated with slower cognitive decline within each level of genetic risk, and the high genetic risk of accelerated cognitive decline was attenuated by a healthier lifestyle, highlighting the importance of engaging in a more favorable lifestyle in the primary prevention of cognitive decline among older adults.

## Methods
### Study design and population
This study was performed within CLHLS, an ongoing prospective population-based cohort study conducted in half of the countries and cities in 23 provinces in China and designed to investigate the determinants of healthy aging among Chinese older adults. Further details about the study design have been previously described elsewhere[42]. In brief, CLHLS was established in 1998 and conducted follow-up surveys and further recruitments of new participants in 2000, 2002, 2005, 2008–2009, 2011–2012, 2014, and 2018. The CLHLS received approval from the Ethics Committee of Peking University (IRB00001052-13074). Written informed consent was obtained from all participants or their legal representatives during the face-to-face interview.

Of the total participants recruited from 8 waves of CLHLS, 37,456 participants completed the baseline and follow-up examinations. We further excluded participants with missing information on lifestyle score at baseline ($n = 671$), those aged less than 65 years old ($n = 297$), those with cognitive impairment at baseline ($n = 9649$), and those without available MMSE measurements at baseline or any follow-up surveys ($n = 8028$). Finally, 18,811 participants remained in the current analysis, and of those, 6301 participants with genotyping data were included for the joint association between healthy lifestyle and genetic risk analysis (Supplementary Fig. 6).

### Lifestyle factors
Data on demographic characteristics, socioeconomic status, lifestyle factors, physical health, and psychological well-being were assessed by well-trained interviewers via the standardized questionnaire at baseline. We derived a healthy lifestyle score based on four modifiable lifestyle factors associated with cognitive function according to prior evidence[30,38,39,43,44], including current non-smoking, never alcohol drinking, active physical activity, and healthy diet intake.

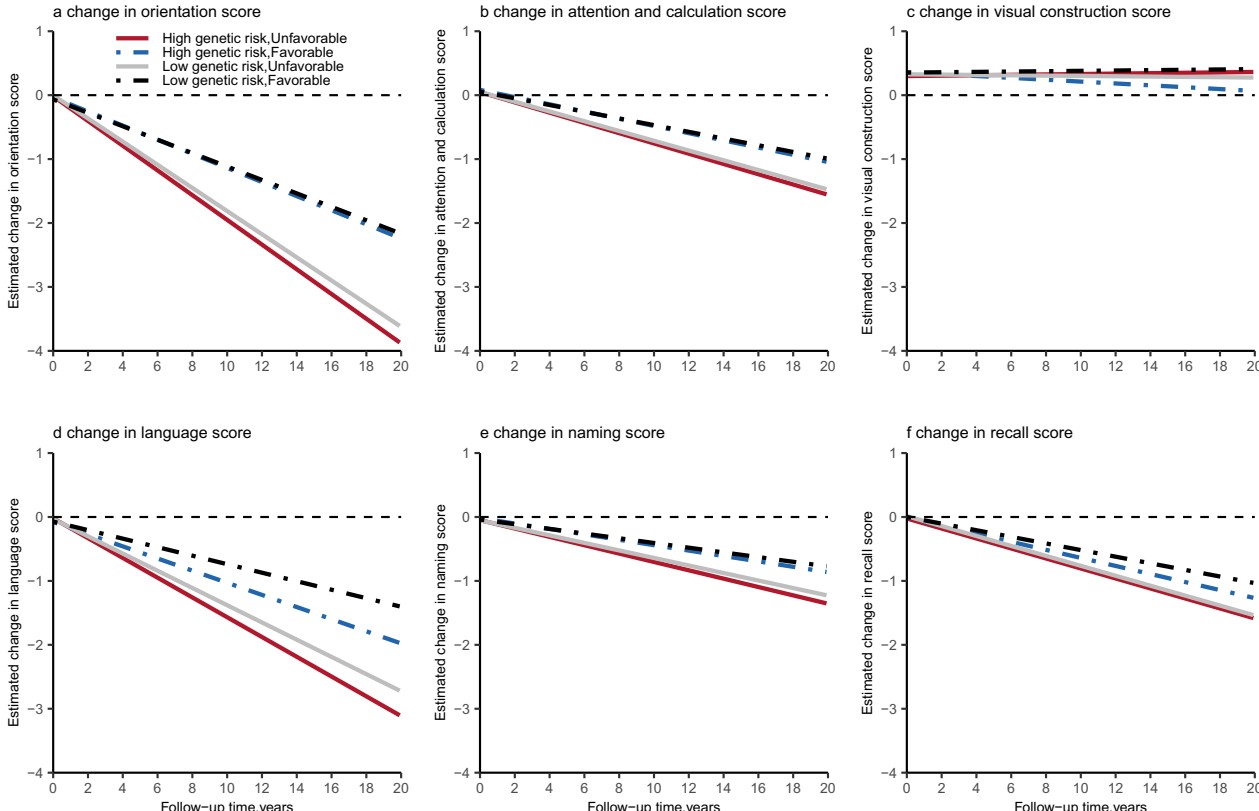

**Fig. 2 | Rate of change in different dimensions of cognitive function according to genetic risk and lifestyle profiles.** Linear mixed-effects models were used with adjustment for age, sex, entry time, educational attainment, area of residence, current marital status, occupation, source of income, and baseline dimensions of cognitive score as appropriate. **a** Joint effect of lifestyle and genetic risk on change in orientation score, **b** joint effect of lifestyle and genetic risk on change in attention and calculation score, **c** joint effect of lifestyle and genetic risk on change in visual construction score, **d** joint effect of lifestyle and genetic risk on change in language score, **e** joint effect of lifestyle and genetic risk on change in naming score, **f** joint effect of lifestyle and genetic risk on change in recall score. Solid red line represents point estimates of high genetic risk and unfavorable lifestyle group, dashed blue line represents point estimates of high genetic risk and favorable lifestyle group, solid gray line represents point estimates of low genetic risk and unfavorable lifestyle group, dashed black line represents point estimates of low genetic risk and favorable lifestyle group. Source data are provided as a Source Data file.

Supplemental methods and Supplementary Table 6 provide additional details on the specific questions asked and the construction of healthy lifestyle scores.

Specifically, current non-smoking was referred to as never smoking or former smoking according to previous studies[15]. Alcohol drinking status was classified as non-drinker, current drinker, and former drinker. The category of never drinking was deemed a healthy lifestyle factor[30]. For physical activity, participants reported the frequency of participation in regular exercises, housework tasks, personal outdoor activities, gardening, rearing domestic animals/pets, reading, playing cards/mahjong, watching TV/listing to the radio, and attending social activities[45]. The frequency of "almost every day", "occasionally", and "rarely or never" was scored 2, 1, or 0, respectively. A total physical activity score was calculated as the sum of 9 activities, which is scored from 0 to 18 with a higher score indicating a greater level of physical activity. An ideal physical activity was defined as a physical activity score in the top 40% of cohort distribution[46]. Dietary intake was assessed by using a standardized food frequency questionnaire with acceptable reproducibility and validity[7], including 9 commonly consumed food groups in the Chinese diet: fresh vegetables, fresh fruit, legumes, meat, eggs, fish and seafood, salty vegetables, tea, and garlic[7,47]. Insufficient total daily protein intake and imbalance in protein synthesis and degradation were observed with aging among older adults, especially the oldest old. Thus, consumption of typical protein-rich food such as legumes, meat, eggs, and fish, which have prominent

beneficial effects on mortality, was also considered as a healthy lifestyle in this study[37,48]. A total diet score was computed in the same way as physical activity. The diet intake was deemed ideal based on the top 40% of the cohort distribution in line with previous studies[46]. All component scores were summed to obtain the healthy lifestyle score ranging from 0 to 4, with a higher index indicating a healthier lifestyle, and were further categorized as "unfavorable" (less than three healthy lifestyle factors), "intermediate" (three healthy lifestyle factors), and "favorable" (four healthy lifestyle factors).

## Covariates
Covariates were selected based on prior research and available cohort measures[15,49]. The Supplementary Information describes covariate details inquired by questionnaires, including age, sex, educational level, area of residence, marital status, occupation, source of income, self-reported health status, optimism status, and history of major chronic disease. Optimism status was evaluated with seven items (score range, 0–7, with higher scores denoting a higher level of optimism)[50]. Presence of major chronic diseases, including cardiovascular disease, diabetes mellitus, hypertension, respiratory disease, digestive system disease, or cancer, was self-reported.

## Cognitive function
Cognitive function was assessed by an adapted Chinese version of the MMSE at baseline and during each follow-up survey according to the

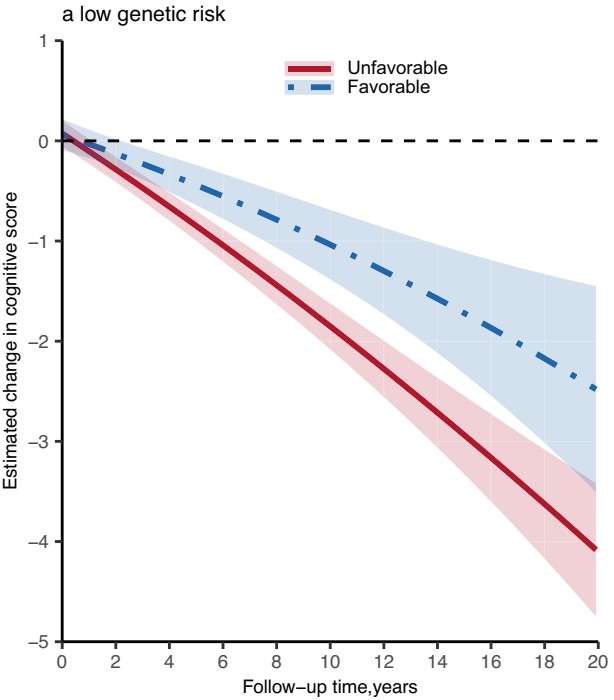

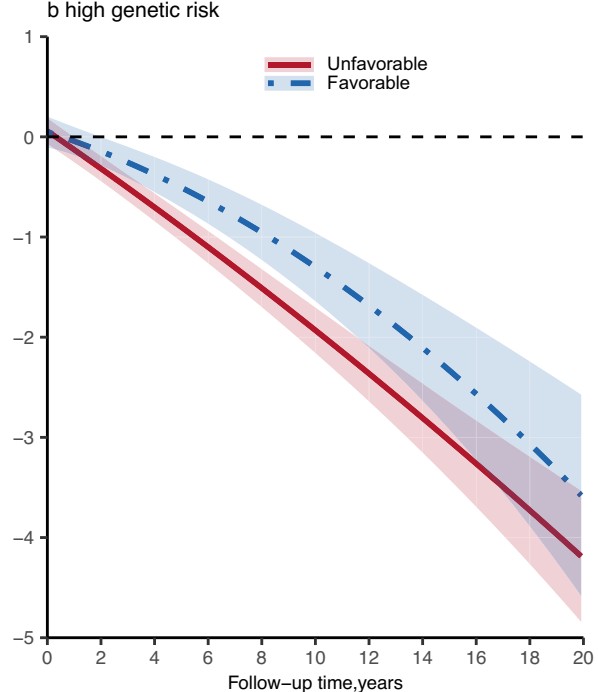

**Fig. 3 | Predicted non-linear change in cognitive function according to genetic risk and lifestyle profiles.** Linear mixed-effects models were used with adjustment for age, sex, entry time, educational attainment, area of residence, current marital status, occupation, source of income, and baseline cognitive score. Potential non-linear relationship was estimated by adding quadratic terms of time to the multivariate model. **a** Among participants with low genetic risk, the non-linear association of lifestyle with cognitive decline, **b** among participants with high genetic risk, the non-linear association of lifestyle with cognitive decline. Solid red line represents point estimates of unfavorable lifestyle group, dashed blue line represents point estimates of favorable lifestyle group, and shaded areas show a 95% confidence interval (CI). Source data are provided as a Source Data file.

standard protocol[29]. The MMSE included 24 items regarding six cognitive dimensions: orientation, attention and calculation, visual construction, language, naming, and recall skills (Supplementary Table 7)[29]. A score of zero was given for incorrect and unknown answers, and one point was given for correct answers. All questions were of equal weight, culminating in a total possible score of 30. Cognitive score and its individual dimensions were transformed from raw score to $Z$ score using the means and SDs at baseline[15]. A positive $Z$ score indicates better cognitive function than the mean population score, and a negative score indicates a poor cognitive score. Higher cognitive score indicates better performance. Cognitive score was computed on each cycle for all study participants, and based on the cognitive score during the follow-up, we determined the rate of cognitive decline. Due to a high proportion of the study participants not having a formal education, cognitive impairment was defined as an overall cognitive score of <18 according to previous studies[29].

### Genotyping and genetic risk score calculation

According to the results of the CLHLS genome-wide association study on longevity, a replication study was carried out among 13,228 individuals using a well-designed and customized chip targeting 27,656 single nucleotide polymorphisms (SNPs) previously associated with longevity and its related traits[51]. Of these selected SNPs, 3966 SNPs were associated with common diseases of old age, i.e., Alzheimer's disease, cardiovascular disease, type 2 diabetes mellitus, cancer, and immune-related diseases. Detailed information on SNPs selection and genotyping process used in the CLHLS study has been published previously[51,52]. The polygenic risk score in the present study was calculated based on the number of risk alleles of 34 genetic variants increasing the risk for Alzheimer's disease as previously published[51,53]. Details regarding the selected SNPs are provided in Supplementary Table 8. Individual SNPs were coded as 0, 1, and 2 according to the

number of risk alleles. The weight coefficient for each SNP was reported in previous studies[51,52]. The genetic risk score was formulated as the sum of the number of risk alleles at each locus multiplied by the respective weight coefficient. The genetic risk score was categorized into two groups according to median: low and high genetic risk groups.

### Statistical analyses

Baseline characteristics of participants were described as mean values (standard deviation) for continuous variables and numbers (percentage) for categorical variables. The marginal mean values for cognitive score and its individual dimensions according to lifestyle categories were estimated by multiple linear regression models with adjustment for baseline covariates.

Linear mixed-effects models were used to test the association of a healthy lifestyle with longitudinal change in cognitive function and each cognitive domain separately. To account for within-individual associations between repeated measurements, all models included time since baseline as well as random intercepts and random slopes of time. Analysis was adjusted for age (years, continuous), sex (female/male), entry time, educational attainment (years of schooling completed <1 year, 1–6 years, or >6 years), area of residence (urban/rural), current marital status (in marriage, not in marriage), occupation (agriculture/forestry/husbandry/fishery, commercial, service, industrial worker/self-employer, professional/governmental/managerial personnel, or houseworker/never worked/other), source of income (independent, dependent), and baseline cognitive score (continuous). Same method was used to evaluate the association between genetic risk and the rate of cognitive decline. We further assessed the association of a healthy lifestyle with cognitive decline stratified by genetic risk, and the interaction between a healthy lifestyle and genetic risk was also evaluated. Additionally, the joint associations of healthy

lifestyle and genetic risk with the rate of cognitive decline. Participants were classified into six categories according to lifestyle group (unfavorable, intermediate, and favorable) and genetic risk (low, high), with the combination of high genetic risk and unfavorable lifestyle group as the reference.

With time-on-study as timescale, person-years were calculated for each participant from baseline to the date of cognitive impairment, death, or end of follow-up, whichever came first. Cox proportional hazard models were used to estimate the associations of lifestyle and genetic factors with incident cognitive impairment, adjusted for age, sex, entry time, educational attainment, area of residence, current marital status, occupation, and source of income. The proportional hazards assumption was examined by the Schoenfeld residuals method, and no violation was identified ($P > 0.05$). We also assessed the influence of death as a competing risk for cognitive impairment via competing risk analyses.

To test the robustness of our results, we performed several sensitivity analyses. First, in addition to primary adjustment, we further controlled for self-reported health status (yes/no), optimism status (continuous), and history of chronic disease status (yes/no) in the multivariate mixed model. Because these factors were identified as risk factors for cognitive function[15]. Second, to explore the possibility of reverse causation due to impaired cognitive function, which might influence the accuracy of reported lifestyle behaviors, participants with baseline MMSE scores below the 10th percentile were excluded[34]. Additionally, changes in cognitive score were truncated at the 0.5th and 99.5th percentiles to minimize the influence of outliers. Third, to address the adverse effects of former smoking, we created a new healthy lifestyle score using "never smoking" as a healthy lifestyle factor. Fourth, due to the common occurrence of missing values for specific items of the MMSE test, such as visual construction, the analysis was restricted to participants who completed all the MMSE items[41]. Fifth, as lifestyle scores might change during follow-up, the latent class trajectory model was used to identify distinct lifestyle score trajectories, and then to assess the effect of lifestyle trajectory groups on cognitive decline. Finally, to account for the non-linear relationship, quadratic terms of time were included in the multivariate model[15]. Two-sided $P < 0.05$ was accepted as statistically significant, except for separate analyses for individual domains of cognition in which the Bonferroni correction was applied to account for multiple testing ($P < 0.008$ considered significant [=0.05/6]). All data analyses were performed in SAS V.9.4 (SAS Institute, Cary, NC, USA), and R V.4.3.1 (R Foundation for Statistical Computing, Vienna, Austria).

### Reporting summary
Further information on research design is available in the Nature Portfolio Reporting Summary linked to this article.

## Data availability
All CLHLS data used in this study, except for the raw genetic sequencing data, are available from the official repository located at https://doi.org/10.18170/DVN/WBO7LK. The raw genetic sequencing data of CLHLS used in this study have been deposited and controlled at the CNGB Sequence Archive (CNSA, https://db.cngb.org/cnsa/) of the China National GeneBank DataBase (CNGBdb) under accession code CNP0000792. The raw genetic sequencing data used in this study are available under restricted access due to data privacy laws and ethical restrictions. Access can be obtained by completing the application form via https://db.cngb.org/data_access/ or by contacting the corresponding authors (shixm@chinacdc.cn or lvyuebin@nieh.chinacdc.cn), and will be answered within 10 weeks. The sharing requests for raw genetic sequencing data of CLHLS will be reviewed by the institutional ethical committees of the CLHLS study to verify whether the request is subject to any intellectual property or confidentiality obligations. This study did not generate new genetic sequencing data.

All other data supporting the findings described in this study are available in the article and the Supplementary Information, Supplementary Data and Source Data files. Source data are provided with this paper.

## Code availability
Code used in this study is available online at https://github.com/wj2016a/hlsprs.git. The DOI for the GitHub repository is https://doi.org/10.5281/zenodo.14135441.

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

## Acknowledgements

We thank all investigators and participants who conducted and participated in the survey. This study was supported by the National Natural Sciences Foundation of China (No. 82222063, Y.L.; 82025030, X.S.; 82388102, H.S.), the National Key Research and Development Program of China (No. 2023YFC3603400, Y.L.), and Young Scholar Science Foundation of China CDC (No. 2022A303, J.W.). The sources of funding had no role in study design, data collection, analyses, interpretation, and decision to submit the article for publication.

## Author contributions

Y.L. and X.S. conceived and designed the study. J.W. performed the statistical analysis, data interpretation, and drafted the manuscript. C.C., J.Z., Z.X., L.X., X.L., and Z.Z. reviewed and edited the manuscript. Y.L. and X.S. had full access to all the data in the study and took responsibility for the integrity of the data and the accuracy of the data analysis. All authors contributed to the manuscript and gave their approval for its final version.

## Competing interests

The authors declare no competing interests.
