## [Transparent Peer Review file · Nature Communications]

Integrated healthy lifestyle even in late-life mitigates cognitive decline risk across varied genetic susceptibility

Corresponding Author: Professor Xiaoming Shi

Version 0:

Reviewer comments:

Reviewer #1

(Remarks to the Author)

This is an interesting study, originating in China and bringing in important information from a relatively old population (mean age 81-82 years). I do think the information is of interest, but I find the innovative character of limited interest, provided the literature on this topic that has already been published. I do want to provide a few suggestions to consider which may help to further improve their important work. The nature and baseline demographics of this population also raise questions. For instance, it is questionable whether lifestyle changes still have beneficial effects at this age. Additionally, it seems questionable how such a relatively old population can be followed for more than 10-20 years, and that also raises the question how reliable the effect estimates will be at later time points. It also for this reason that I missing a confidence interval in figure 1a.

Other major points of concern:

- 1) The abstract implies an intervention study, where individuals shift from an unhealthy lifestyle to a healthier one ('adopting'). Since this is an observational study without target trial emulation techniques, this claim does not seem to hold and I would advise to change wording.
- 2) The population is of interest, it seems that a lot of people did not went to school, since more than half of the population has less than 1 year (!) of education. Does this refer to/include high school only or also elementary levels?
- 3) Occupation status is others for 91% of the population, since this is such a large proportion, it warrants more information that only mentioning 'others'
- 4) The genetic analyses seem only modestly related to cognition, whereas lifestyle factors appear to have a stronger association. It would be helpful to readers to provide tables where genetic information is related to cognition, without considering lifestyle factors, in order to assess the independent strength of the analyses compared to lifestyle factors in relation to cognition.
- 5) In addition to point 2, I see a GRS based on SNPs, but I miss important genetic risk factors like APOE. Did the authors consider to include this in their analyses?
- 6) The authors extensively adjusted for confounders, and even performed several sensitivity analyses including self-reported measures of health.
- 7) The authors provide access to their datasets and code.
- 8) Surprising to see that there is little change in the visual spatial domain for cognition, this intrigues me since patients at risk for dementia generally miss out points on this domain in an early phase of the disease.
- 9)

Minor points:

Figure 1 is insightful, but also raises questions. Why do all participants start at the same intercept ? would it not be more in line with biology to allow for random intercepts, in order to capture differences at baseline at the advanced age of 81?

Note that the authors selected individuals that completed all cognitive tests/visits, this could lead to selection bias. It would be helpful to comment on this matter, and to quantify whether this could have affected results.

With this large population it seems to be possible to account for non-linearity, did the authors consider to fit in natural splines, to account for non-linear changes of cognition at this age?

(Remarks on code availability)

the code is brief, and does not seem to capture all steps in the analysis process and requires a bit more read.me information to follow

Reviewer #2

(Remarks to the Author)

This study highlights important aspect of cognitive trajectory based on lifestyle and genetic risk subgroups in a longitudinal population-based cohort from the Chinese descent. Study design and methods are extensive and reasonable, while results are convincing. However, the manuscript was missing with some details or misplaced due to the Journal's format, which need to be improved.

Methods and Results:

Lines 103-104: In the Results section, please include summary of lifestyle (Unfavorable/Intermediate/Favorable) and genetic (low vs high genetic risk) subgroups from Methods since this description came in Methods.

What is the rationale including SNPs associated other diseases?

Line 139: Subtitles seemed grammatically wrong. Suggest revising them.

Table S6: % difference calculation did not consider point reference's point estimate. For example, visual construction score difference was 100% between favorable and unfavorable. However, estimate was small (-0.006 vs. -0.001). Interpretation should be cautious.

Genetic risk groups: how about just using APOE e4 carrier status. Was PRS based subgrouping better?

Figure 2: It is unclear about the joint model including both lifestyle and genetic profiles improved the overall prediction on cognitive decline.

Line 163: this analysis seemed related to progression to cognitive impairment. Subheading was misleading as a cross-sectional analysis. Suggest changing it.

Lines 178-181: Sensitivity analysis tables were very difficult to follow. Any significant results? Please summarize key findings with p values.

Discussion:

Lines 180-192: "Healthy lifestyle might outweigh genetic background in relation to cognitive decline". This statement may not be fair. Since, Also, genetic risk subgroups are dependent on the list of risk SNPs and distribution of the PRS. This study used SNPs associated with other diseases and used median of PRS distribution for subgrouping. If you choose AD associated SNPs and more extremes of the PRS distribution using the AD associated, you may see larger % differences between genetic subgroups.

APOE e4 vs. PRS: the current study did not show whether subgrouping with PRS approach is better than that with the APOE e4 allele alone.

(Remarks on code availability)

Reviewer #3

(Remarks to the Author)

I think this is an important study, re-confirming the role of lifestyle on cognition and showing that a healthy lifestyle may attenuate the genetic risk.

Overall, it is a well-conducted study, and I have only a few comments.

1. Does the lifestyle change during the follow-up? I suggest showing a figure for trajectories of lifestyle scores during the follow-up.

2. MMSE is limited in determining the cognitive impairment or evaluating the change in those with poor cognition. Maybe sensitivity analysis in which authors exclude the bottom 10% of people with low scores?

3. When investigating each domain, e.g., visual, language, etc., it may require p-value adjustment for multiple testing.

4. The genetic risk seems to have a negligible effect on cognition, only 0.007 SDU. A comment in a discussion comparing to other studies in the literature is necessary.

(Remarks on code availability)

Version 1:

Reviewer comments:

Reviewer #1

(Remarks to the Author)

Thank you for your extensive and clear comments. These have definitively contributed to improve the quality of the paper. I would strongly advice to use your non-linear version of Figure R1-2 in the main paper, while including confidence intervals. I particularly value your efforts to include a systematic review to stress the novelty of their findings. Thanks you for the opportunity to review this paper.

(Remarks on code availability)

Reviewer #3

(Remarks to the Author)

Thank you for addressing my comments. No additional comments.

(Remarks on code availability)

Responses to reviewers' comments

Dear reviewers of Nature Communications:

Thank you for your detailed informative comments and suggestions which helped to improve our manuscript entitled “Integrated healthy lifestyle even in late-life mitigates cognitive decline risk across varied genetic susceptibility” (NCOMMS-24-00647-A). We have made revisions according to your comments and provided a detailed point-by-point response. We have numbered the comments for easy reference as R1-1, R1-2, ... for the first reviewer, R2-1, R2-2, ... for the second reviewer, R3-1, R3-2, ... for the third reviewer. Enclosed please find the responses and the revised manuscript with all the modifications highlighted in yellow. We greatly appreciate your reconsideration of our manuscript.

Comments of Reviewer 1

R1-1 This is an interesting study, originating in China and bringing in important information from a relatively old population (mean age 81-82 years). I do think the information is of interest, but I find the innovative character of limited interest, provided the literature on this topic that has already been published. I do want to provide a few suggestions to consider which may help to further improve their important work.

Response: Thanks for your valuable comments and constructive suggestions, which were very helpful in improving our manuscript.

- (1) As suggested, a comprehensive understanding of implementation and data quality of the Chinese Longitudinal Healthy Longevity Survey (CLHLS) study have been provided in detail, a polygenic risk score (PRS) using 34 SNPs associated with Alzheimer's disease (AD) risk has been constructed to estimate the independent effect of genetic risk on cognitive decline, and the annotated SAS code, corresponding datasets for data analysis has been provided and updated.
- (2) Comparing with the prior studies, we further investigate the effect of combined genetic and lifestyle factors on subsequent cognitive decline and its six specific dimensions using data from the CLHLS, including approximately two-thirds of oldest-old, and the findings highlight the importance of adhering to favorable lifestyle for a better cognitive function regardless of current cognitive status and genetic risk among older adults (Table R1-1).
- (3) Based on the suggestions, the following major changes have been made to improve our study:
 - To elaborate the nature and baseline demographics of included participants, such as educational level, occupation status, and follow-up time duration, a comprehensive understanding of implementation and data quality of CLHLS study have been provided in detail.
 - To clearly explain the details of included 34 SNPs in the construction of the weighted PRS, and the independent impact of genetic risk on cognitive decline, detailed information and further analysis have been added.

- To capture all steps in the data analysis process and to maximize the reproducibility of the results, the annotated SAS code, corresponding datasets for data analysis, and a read-me file have been provided and submitted to GitHub.
 - To more objectively evaluate the association of combined genetic and lifestyle factors on subsequent cognitive decline and its specific dimensions, and clearly identify the applicability and generalizability of our findings, the following points has been acknowledged in a new limitation in the revised manuscript, which provide valuable insights for future research to expand upon these findings and further enhance our understanding of the topic:
 - ✓ due to the relatively lower educational level of including participants, the generalizability of our findings to those with higher levels of education was limited;
 - ✓ due to limited number of items to assess the visual construction and common occurrence of missing values in this domain, nonsignificant association between healthy lifestyle and genetic risk on visual construction should be interpreted with caution;
 - ✓ due to the limited number of including variants associated with cognitive decline for construction of PRS, further studies were warranted by comprehensive coverage of established cognitive decline related genetic variants to construct PRS, and improve the accuracy of risk classification.
- (4) A systematic literature review focus on the association of healthy lifestyle, genetic risk, and cognitive decline has been conducted to identify the added value of present analysis compared with previous studies as follows (Table R1-1).

Table R1-1. Research in context and added value of this study*

	Evidence before this study	Gaps	Added value of this study
Lifestyle factors	Focus on the impact of individual lifestyle factors for overall cognitive decline, with conflicting results (Ref:1-5)	Scarce evidence about combined lifestyle on rate of cognitive decline and its specific dimensions (Ref:1-5)	Investigating the combined lifestyle with rate of cognitive decline and specific dimensions of cognition, and identifying which dimensions are most affected
Genetic risk factors	Most studies reporting the interaction between lifestyle, both individually and in combinations, and APOE ε4 on cognitive decline (Ref:1,6-7)	Few studies investigating the association of combined lifestyle and weighted polygenic risk (combination of multiple individual SNP effects) on rate of cognitive decline (Ref:8-9)	Exploring the association of combined lifestyle and weighted polygenic risk on rate of cognitive decline and specific dimensions of cognition
Outcome	Focus on risk of incident cognitive impairment or dementia, or rate of overall cognitive decline (Ref:1,6-7,10-11)	Limited evidence about rate of overall cognitive decline and specific dimensions of cognition	
Population	Focus on individuals between 60 and 80 years of age (Ref:1)	Insufficient evidence for an association between combined lifestyle and cognitive decline in Chinese oldest-old	Using data from the Chinese Longitudinal Healthy Longevity Survey (CLHLS), including approximately two-thirds of oldest-old
Implications	 ● Adherence to a favorable lifestyle was associated with slower decline in cognitive function and five specific domains of cognition, such as orientation, attention and calculation, language, naming, and recall. 		

- Individuals with high genetic risk were more likely to experience cognitive decline compared with those with low genetic risk.
- Adherence to a favorable lifestyle has been related to a slower cognitive decline even in those with high genetic risk.
- A favorable lifestyle might outweigh genetic background in delaying cognitive decline, highlighting the importance of adhering to favorable lifestyle for a better cognitive function regardless of current cognitive status and genetic risk among older adults.

* We systematically searched PubMed and Google Scholar using the search terms “lifestyle” AND (“genetic risk” OR “genetic” OR “genetic factors” OR “APOE”) AND (“cognitive decline” OR “cognitive function” OR “cognitive health” OR “cognitive performance” OR “cognitive change” OR “cognitive ability”) for studies published from inception to April 1, 2024, with no language restrictions.

Reference:

1. Ye KX, et al. *The role of lifestyle factors in cognitive health and dementia in oldest-old: A systematic review. Neurosci Biobehav Rev* 152, 105286 (2023).
2. Kheirouri S, Alizadeh M. *MIND diet and cognitive performance in older adults: a systematic review. Critical reviews in food science and nutrition* 62, 8059-8077 (2022).
3. Iso-Markku P, et al. *Physical Activity and Cognitive Decline Among Older Adults: A Systematic Review and Meta-Analysis. JAMA network open* 7, e2354285 (2024).
4. Dominguez LJ, et al. *Nutrition, Physical Activity, and Other Lifestyle Factors in the Prevention of Cognitive Decline and Dementia. Nutrients* 13, 4080 (2021).
5. Ding Z, Leung PY, Lee TL, Chan AS. *Effectiveness of lifestyle medicine on cognitive functions in mild cognitive impairments and dementia: A systematic review on randomized controlled trials. Ageing research reviews* 86, 101886 (2023).
6. Dhana K, et al. *Genetic risk, adherence to a healthy lifestyle, and cognitive decline in African Americans and European Americans. Alzheimers Dement* 18, 572-580 (2022).
7. Dhana K, Aggarwal NT, Rajan KB, Barnes LL, Evans DA, Morris MC. *Impact of the Apolipoprotein E epsilon4 Allele on the Relationship Between Healthy Lifestyle and Cognitive Decline: A Population-Based Study. Am J Epidemiol* 190, 1225-1233 (2021).
8. Fan J, et al. *The Contribution of Genetic Factors to Cognitive Impairment and Dementia: Apolipoprotein E Gene, Gene Interactions, and Polygenic Risk. Int J Mol Sci* 20, 1177 (2019).
9. Pan G, et al. *The potential roles of genetic factors in predicting ageing-related cognitive change and Alzheimer's disease. Ageing research reviews* 70, 101402 (2021).
10. Jin X, et al. *Association of APOE ε4 genotype and lifestyle with cognitive function among Chinese adults aged 80 years and older: A cross-sectional study. PLoS Med* 18, e1003597 (2021).
11. Duan H, et al. *Association of Unhealthy Lifestyle and Genetic Risk Factors With Mild Cognitive Impairment in Chinese Older Adults. JAMA network open* 6, e2324031 (2023).

R1-2 The nature and baseline demographics of this population also raise questions. For instance, it is questionable whether lifestyle changes still have beneficial effects at this age.

Response: Thanks for you pointing this out. Benefits of adhering to healthy lifestyle for lifetime gain or dementia prevention persist in older individuals, albeit to a lesser extent than younger adults, and long-term health benefits of engaging in a healthy lifestyle showed downward trends with age (*Ref:1-4*).

Besides, more evidence focused on the effect of lifestyle, individually or in combination, on cognitive health in younger elderly, with insufficient attention paid to explore the impact of combined lifestyle on rate of cognitive decline among oldest-old during the last stage of their lifecycle (*Ref:5-6*). Based on CLHLS data, including approximately two-thirds of oldest-old, we observed that adherence to favorable lifestyle was associated with slower rate of cognitive decline even in older adults at high genetic risk.

Additionally, we have discussed this issue more deeply in the revised manuscript (*Discussion section, Page 9, Lines 263-272 in the revised manuscript*): “Moreover, the study sample comprised predominantly oldest-old, and the benefits of adhering to healthy lifestyle for better cognitive function persist in oldest-old. These findings align with a latest study showing that a healthy lifestyle was associated with better cognitive function proximate to death, with an average death age of 90 years, independently of common neuropathologies of dementia³⁵. A recent systematic review found that individual lifestyle factors could lower the risk of developing cognitive impairment and dementia even among oldest-old, while the evidence examining the effect of combined lifestyle on rate of cognitive decline in oldest-old is insufficient⁹.”

Reference:

1. Sun Q, et al. Healthy lifestyle and life expectancy at age 30 years in the Chinese population: an observational study. *Lancet Public Health* 7, e994-e1004 (2022).
2. Wang J, et al. Healthy lifestyle in late-life, longevity genes, and life expectancy among older adults: a 20-year, population-based, prospective cohort study. *Lancet Healthy Longev* 4, e535-e543 (2023).
3. Rizzuto D, Orsini N, Qiu C, Wang HX, Fratiglioni L. Lifestyle, social factors, and survival after age 75: population based study. *BMJ (Clinical research ed)* 345, e5568 (2012).
4. Livingston G, et al. Dementia prevention, intervention, and care: 2020 report of the Lancet Commission. *Lancet (London, England)* 396, 413-446 (2020).
5. Ye KX, et al. The role of lifestyle factors in cognitive health and dementia in oldest-old: A systematic review. *Neurosci Biobehav Rev* 152, 105286 (2023).
6. Ding Z, Leung PY, Lee TL, Chan AS. Effectiveness of lifestyle medicine on cognitive functions in mild cognitive impairments and dementia: A systematic review on randomized controlled trials. *Ageing research reviews* 86, 101886 (2023).

R1-3 Additionally, it seems questionable how such a relatively old population can be followed for more than 10-20 years, and that also raises the question how reliable the effect estimates will be at later time points.

Response: Thanks for your advice. First, the successful implementation of each follow-up survey of CLHLS study, and acquisition of high-quality data could be attributable to the continuous financial support from domestic and international, systematic work networks for field investigation, and the relatively low migration rate of Chinese older adults. Second, to explore the reliability about the effect estimates at later time points, we further conducted association analysis about healthy lifestyle with rate of cognitive decline stratified by duration of follow-up time (Tabel R1-2), and the magnitude of the benefits of adhering to a healthy lifestyle was consistent with the main analysis.

- (1) The CLHLS survey has been long-term supported by NIA/NIH, UNFPA, China Natural Sciences Foundation (NSFC), and other Chinese resources (Ref:1-4). The design of the CLHLS questionnaire is based on international standards and adapted to the Chinese cultural/social context and carefully tested by pilot studies/interviews (Ref:1-3). Strict data quality assessment is conducted at each wave survey in terms of nonresponse rate, sample attrition, logical errors, and reliability and validity of the major health measures (Ref:1-5). The CLHLS data has been shared within institutions worldwide to conduct scientific studies and analyses among Chinese older adults.
- (2) The Chinese Center for Disease Control and Prevention (CDC) has undertaken the investigation of CLHLS since 2008 (Ref:6-7). The CDC network, as a multi-collaboration system, ensure the orderly conduct of follow-up survey, and the quality of investigation. China CDC performs overall project review and validation. The provincial CDC provides technical guidance and participate in field quality control. Health clinics in towns and townships, community health service centers and village medical personnel also involve in the research work (Ref:6-7).
- (3) The scale of older adults' migration in China is notably smaller than that of younger people (Ref:8-9). Approximately 6.6% of the Chinese adults aged 60 and older migrated in the past 10 years (Ref:8-9). The low migration rate among older adults ensures that a high proportion of study participants could be followed.
- (4) The Tabel R1-2 shows that individuals with follow-up time of less than 10 years had a 44.03% slower decline in favorable lifestyle group than in the unfavorable lifestyle group. Among participants with follow-up time of 10 years or above, those with favorable lifestyle had a 30.77% slower decline than those with unfavorable lifestyle.

Table R1-2. Association of healthy lifestyle with rate of cognitive decline stratified by duration of follow-up time

Lifestyle	Standard deviation units			Difference, %
	Estimate (95% CI)	Difference (95% CI)	P value	
Follow-up time of less than 10 years (N=15901)				
Unfavorable	-0.477 (-0.494, -0.460)	Reference		Reference
Intermediate	-0.420 (-0.443, -0.397)	0.057 (0.029, 0.086)	0.0001	11.95
Favorable	-0.267 (-0.303, -0.232)	0.210 (0.171, 0.249)	<0.0001	44.03
Follow-up time of 10 years or above (N=2901)				
Unfavorable	-0.104 (-0.115, -0.092)	Reference		Reference

Intermediate	-0.088 (-0.100, -0.075)	0.016 (0.000, 0.032)	0.0487	15.38
Favorable	-0.071 (-0.089, -0.054)	0.032 (0.013, 0.052)	0.0014	30.77

Models were adjusted for age, sex, entry time, educational attainment, area of residence, current marital status, occupation, source of income, and baseline cognitive score.

Reference:

1. Yi Z, Poston D, Vlosky DA, Gu D. *Healthy Longevity in China: Demographic, Socioeconomic, and Psychological Dimensions*. Springer Publisher (2008).
2. Zeng Y, Feng Q, Hesketh T, Christensen K, Vaupel JW. *Survival, disabilities in activities of daily living, and physical and cognitive functioning among the oldest-old in China: a cohort study*. *Lancet (London, England)* 389, 1619-1629 (2017).
3. Zheng Z, Shi X, Zeng Y, Lei X. *Chinese Longitudinal Healthy Longevity Survey and Database Construction*. Longmen Publishing House (2021).
4. Duke Aging Center. *Chinese Longitudinal Healthy Longevity Study*. [cited 2024 04.10] Available from: <https://agingcenter.duke.edu/CLHLS>.
5. Lv YB, et al. *Revisiting the association of blood pressure with mortality in oldest old people in China: community based, longitudinal prospective study*. *BMJ (Clinical research ed)* 361, k2158 (2018).
6. Lv Y, Mao C, Yin Z, Li F, Wu X, Shi X. *Healthy Ageing and Biomarkers Cohort Study (HABCS): a cohort profile*. *BMJ open* 9, e026513 (2019).
7. Zeng Y. *Towards Deeper Research and Better Policy for Healthy Aging --Using the Unique Data of Chinese Longitudinal Healthy Longevity Survey*. *China Economic J* 5, 131-149 (2012).
8. Dou X, Liu Y. *Elderly Migration in China: Types, Patterns, and Determinants*. *J Appl Gerontol* 36, 751-771 (2017).
9. Qin B, Peng Y, Wan S. *The geography of older adults' migration in China: Spatial patterns and driving forces*. *Population, Space and Place*, (2024). doi:10.1002/psp.2754.

R1-4 It also for this reason that I missing a confidence interval in figure 1a.

Response: Thanks for your advice. As suggested, we have provided the confidence intervals for the joint effect of genetic risk and healthy lifestyle with rate of cognitive decline in **Figure R1-1** and Figure 1a in the revised manuscript.

Figure R1-1. Rate of change in cognitive function according to genetic risk and lifestyle profiles. Models were adjusted for age, sex, entry time, educational attainment, area of residence, current marital status, occupation, source of income, and baseline cognitive score.

R1-5 The abstract implies an intervention study, where individuals shift from an unhealthy lifestyle to a healthier one (‘adopting’). Since this is an observational study without target trial emulation techniques, this claim does not seem to hold and I would advise to change wording.

Response: Thanks for your careful review. As suggested, we have carefully checked throughout the whole text, and replaced the expression “adopting” with more accurate expression of “adhering to” in the revised manuscript. We have modified the Abstract as follows in the revised manuscript (Abstract section, Page 2, Lines 41-56 in the revised manuscript): *“Whether the benefits of adhering to healthy lifestyle outweigh the adverse influence of inferior genetic background on cognitive decline risk is unclear...Individuals with high genetic risk but adhering to a favorable lifestyle have a slower cognitive decline, compared with those with low genetic risk and an unfavorable lifestyle. The benefits of adhering to a favorable lifestyle on cognitive decline is stronger than those of genetic factors, suggesting that maintaining a favorable lifestyle may offset the genetic risk for accelerated cognitive decline.”*

R1-6 The population is of interest, it seems that a lot of people did not went to school, since more than half of the population has less than 1 year (!) of education. Does this refer to/include high school only or also elementary levels?

Response: Thanks for pointing this out. The educational level of the included participants tends to be low (Ref:1). According to the National Population Census Data in 2000, about half of the population aged 65 year or older was illiterate (Ref:2). Two studies, China Health and Retirement Longitudinal Study (CHARLS) and China Health and Nutrition Survey (CHNS), conducted in the same period showed consistent results (Ref:3-4), with nearly half of older adults being illiterate.

We have modified the education category as follows in the revised Supplementary information: *“The educational level was based on self-reported years of schooling and further categorized as “no formal education (educational attainment <1 year)”, “elementary school level (1–6 years of schooling)” or “high school level or above (more than 6 years of schooling)”* (Ref:5). The educational level of the included participants is shown in Table R1-3, and the results have been updated in Table 1 and Supplementary Table S4. We have also added this as a limitation in the revised manuscript (Discussion section, Page 11, Lines 323-325 in the revised manuscript): *“Another significant limitation is that the study sample included primarily individuals with low educational level, thus limit the generalizability of our findings to those with higher levels of education.”*

Table R1-3. The educational level of the included participants

Characteristic	All participants (N=18811)	Participants with genetic information (N=6301)
Educational attainment, N (%)		
Less than 1 year	10612 (56.41)	3331 (52.86)
1–6 years	6043 (32.12)	2153 (34.17)
More than 6 years	2156 (11.46)	817 (12.97)

Number (percentage) for dichotomous variables.

Reference:

1. Jin X, et al. Association of APOE $\epsilon 4$ genotype and lifestyle with cognitive function among Chinese adults aged 80 years and older: A cross-sectional study. *PLoS Med* 18, e1003597 (2021).
2. National Bureau of Statistics of China. Tabulation of the 2000 population census of the People's Republic of China 2000. [cited 2024 04.10] Available from: <https://www.stats.gov.cn/sj/pcsj/rkpc/5rp/index1.htm>.
3. Huang W, Zhou Y. Effects of education on cognition at older ages: evidence from China's Great Famine. *Soc Sci Med* 98, 54-62 (2013).
4. Chen F, Yang Y, Liu G. Social Change and Socioeconomic Disparities in Health over the Life Course in China: A Cohort Analysis. *Am Sociol Rev* 75, 126-150 (2010).
5. Xi D, et al. Risk factors associated with heatwave mortality in Chinese adults over 65 years. *Nat Med*, (2024). doi: 10.1038/s41591-024-02880-4.

R1-7 Occupation status is others for 91% of the population, since this is such a large proportion, it warrants more information that only mentioning 'others'.

Response: Thanks for your constructive suggestions. We have modified the classification of the "occupation status" in the revised Supplementary information, which were classified into four categories (*Ref:1*): "agriculture/forestry/husbandry/fishery", "commercial, service, industrial worker/self-employed", "professional/governmental/managerial personnel", or "houseworker/never worked/other".

Among about 91% of the study population whose occupation status was previously classified as "others", 61.07% were "agriculture/forestry/husbandry/fishery", 17.05% were "commercial, service, or industrial worker/self-employer", and 12.26% were "houseworker/never worked/other". The "occupation status" of the included participants is presented in Table R1-4, and the results have been updated in Table 1 and Supplementary Table S4.

Table R1-4. The occupation status of the included participants

Characteristic	All participants (N=18811)	Participants with genetic information (N=6301)
Occupation status, N (%)		
Agriculture/forestry/husbandry/fishery	11488 (61.07)	4220 (66.97)
Commercial, service, or industrial worker/self-employer	3208 (17.05)	998 (15.84)
Professional/governmental/managerial personnel	1809 (9.62)	606 (9.62)
Houseworker/never worked/other	2306 (12.26)	477 (7.57)

Number (percentage) for dichotomous variables.

Reference:

1. Xi D, et al. Risk factors associated with heatwave mortality in Chinese adults over 65 years. *Nat Med*, (2024). doi: 10.1038/s41591-024-02880-4.

R1-8 The genetic analyses seem only modestly related to cognition, whereas lifestyle factors appear to have a stronger association. It would be helpful to readers to provide tables where genetic information is related to cognition, without considering lifestyle factors, in order to assess the independent strength of the analyses compared to lifestyle factors in relation to cognition.

Response: Thanks for your reminding. All the included 34 SNPs for construction of polygenetic risk score were associated with increased risk of AD according to previous genome-wide association studies (*Ref:1-2*). The association of genetic risk factors alone with rate of cognitive decline has been provided in Table R1-5 and Supplementary Table S5 as suggested.

Participants with low genetic risk showed significantly slower cognitive decline compared with those with high genetic risk (difference, 0.023 SDU per year [95% CI, 0.003–0.044 SDU per year], *P* value=0.0253). The independent effect of low genetic risk on cognitive decline was slightly weaker compared with the impact of favorable lifestyle (difference, 0.176 SDU [95%CI, 0.147–0.205 SDU per year], *P*<0.0001).

As aging is the most substantial factor for AD (*Ref:3-4*) and genetic information tends to remain relatively stable, we previously estimated the association between genetic risk factors and cognitive decline using age as the time-scale (*Ref:5*). To ensure comparability with similar studies about the impact of genetic risk and healthy lifestyle on cognitive decline, follow-up time scale was used to evaluate the genetic and lifestyle factors on cognitive decline during the follow-up in the revised manuscript (*Ref:6-7*).

Table R1-5. Rate of change in cognitive score according to genetic risk

Genetic risk	Standard deviation units			Difference, %
	Estimate (95% CI)	Difference (95% CI)	P value	
Cognitive score				
High genetic risk	–0.184 (–0.199, –0.169)	Reference		Reference
Low genetic risk	–0.161 (–0.176, –0.146)	0.023 (0.003, 0.044)	0.0253	12.50
Orientation score				
High genetic risk	–0.177 (–0.194, –0.160)	Reference		Reference
Low genetic risk	–0.152 (–0.169, –0.135)	0.025 (0.002, 0.049)	0.0359	14.12
Attention and calculation score				
High genetic risk	–0.074 (–0.082, –0.067)	Reference		Reference
Low genetic risk	–0.066 (–0.073, –0.058)	0.009 (–0.002, 0.019)	0.1019	12.16
Visual construction score				
High genetic risk	–0.003 (–0.008, 0.003)	Reference		NA
Low genetic risk	0.000 (–0.005, 0.006)	0.003 (–0.004, 0.011)	0.4058	
Language score				
High genetic risk	–0.137 (–0.150, –0.123)	Reference		Reference
Low genetic risk	–0.113 (–0.127, –0.100)	0.023 (0.005, 0.042)	0.0139	16.79
Naming score				
High genetic risk	–0.059 (–0.067, –0.050)	Reference		Reference
Low genetic risk	–0.048 (–0.056, –0.039)	0.011 (–0.001, 0.023)	0.0630	18.64
Recall score				

High genetic risk	-0.078 (-0.087, -0.069)	Reference	Reference
Low genetic risk	-0.066 (-0.075, -0.058)	0.012 (-0.000, 0.023)	0.0553 15.38

Analyses were adjusted for age, sex, entry time, educational attainment, area of residence, current marital status, occupation, source of income, and baseline cognitive score. For the changes in each of the individual cognitive score, models were additionally adjusted for the baseline dimensions of cognitive score as appropriate instead of baseline cognitive score.

Reference:

1. Liu X, et al. Integrated genetic analyses revealed novel human longevity loci and reduced risks of multiple diseases in a cohort study of 15,651 Chinese individuals. *Aging Cell* 20, e13323 (2021).
2. Desikan RS, et al. Genetic assessment of age-associated Alzheimer disease risk: Development and validation of a polygenic hazard score. *PLoS Med* 14, e1002258 (2017).
3. 2023 Alzheimer's disease facts and figures. *Alzheimers Dement* 19, 1598-1695 (2023).
4. Legdeur N, Heymans MW, Comijs HC, Huisman M, Maier AB, Visser PJ. Age dependency of risk factors for cognitive decline. *BMC geriatrics* 18, 187 (2018).
5. Jia J, et al. Association between healthy lifestyle and memory decline in older adults: 10 year, population based, prospective cohort study. *BMJ (Clinical research ed)* 380, e072691 (2023).
6. Dhana K, et al. Genetic risk, adherence to a healthy lifestyle, and cognitive decline in African Americans and European Americans. *Alzheimers Dement* 18, 572-580 (2022).
7. Dhana K, Aggarwal NT, Rajan KB, Barnes LL, Evans DA, Morris MC. Impact of the Apolipoprotein E epsilon4 Allele on the Relationship Between Healthy Lifestyle and Cognitive Decline: A Population-Based Study. *Am J Epidemiol* 190, 1225-1233 (2021).

R1-9 In addition to point 2, I see a GRS based on SNPs, but I miss important genetic risk factors like APOE. Did the authors consider to include this in their analyses?

Response: Thanks for pointing this out. Apolipoprotein E (*APOE*) was included in our polygenic risk score. The detail information about the included 34 SNPs is present in Table R1-6 and Supplementary Table S3. All the selected SNPs were associated with increased risk of AD according to previous genome-wide association studies (Ref:1-2).

Table R1-6. List of SNPs included in the polygenic risk score

SNP #	Chromosome	Chromosome position	Gene	Risk Allele	Beta	P value
rs6656401	1	207518704	CR1	A	0.1655	6.00E-24
rs6701713	1	207612944	CR1	A	0.1484	5.00E-10
rs11889338	2	17246318	ZFYVE9P2, RN7SKP168	A	0.4383	9.00E-06
rs7561528	2	127132061	NIFKP9, BIN1	A	0.1570	4.00E-14
rs6733839	2	127135234	NIFKP9, BIN1	T	0.1989	7.00E-44
rs1552244	3	10093893	FANCD2, FANCD2OS	A	0.2776	2.00E-06
rs6448799	4	11628425	HS3ST1, LINC02360	T	0.0770	7.00E-08
rs727153	4	154733269	LRAT	C	0.4886	3.00E-06
rs190982	5	88927603	MEF2C-AS1	A	0.0770	3.00E-08
rs9271192	6	32610753	HLA-DRB1, HLA-DQA1	C	0.1044	3.00E-12

rs9381040	6	41186912	TREM2, TREML2	C	0.0770	6.00E-07
rs6922617	6	41368363	NCR2, FOXP4-AS1	A	0.0900	4.00E-08
rs9349407	6	47485642	CD2AP	C	0.1044	9.00E-09
rs10948363	6	47520026	CD2AP	G	0.0953	5.00E-11
rs1476679	7	100406823	ZCWPW1	T	0.0953	6.00E-10
rs11771145	7	143413669	EPHA1-AS1	G	0.1044	1.00E-13
rs10273775	7	147200311	CNTNAP2	G	0.4187	9.00E-06
rs28834970	8	27337604	PTK2B	C	0.0953	7.00E-14
rs9331896	8	27610169	CLU	T	0.1484	3.00E-25
rs7818382	8	95041772	NDUFAF6	T	0.0677	8.00E-08
rs956225	8	121897448	HAS2-AS1	A	1.2030	9.00E-06
rs514716	9	3929424	GLIS3	G	0.0710	3.00E-09
rs7920721	10	11678309	USP6NL-AS1, ECHDC3	G	0.0677	3.00E-07
rs474951	11	60071148	MS4A3, MS4A2	T	0.2390	1.00E-06
rs10792832	11	86156833	RNU6-560P, LINC02695	G	0.1398	9.00E-26
rs17511627	13	26150190	ATP8A2P3, RNF6	C	0.5596	5.00E-06
rs17125944	14	52933911	FERMT2	C	0.1310	8.00E-09
rs10498633	14	92460608	SLC24A4	G	0.0953	6.00E-09
rs3752246	19	1056493	ABCA7	G	0.1398	6.00E-07
rs4147929	19	1063444	ABCA7	A	0.1398	1.00E-15
rs6859	19	44878777	NECTIN2	A	0.3436	1.00E-07
rs8035452	15	50748601	SPPL2A	T	0.0770	3.00E-07
rs429358*	19	44908684	APOE	T		
rs7412*	19	44908822	APOE	C		

#All the selected SNPs were associated with Alzheimer's disease.

*APOE ϵ 2 and ϵ 4 allele were obtained based on the combination of APOE rs429358 and APOE rs7412 polymorphisms, with the effect value for Alzheimer's disease of -0.47 and 1.03, respectively.

Reference:

1. Liu X, et al. Integrated genetic analyses revealed novel human longevity loci and reduced risks of multiple diseases in a cohort study of 15,651 Chinese individuals. *Aging Cell* 20, e13323 (2021).
2. Desikan RS, et al. Genetic assessment of age-associated Alzheimer disease risk: Development and validation of a polygenic hazard score. *PLoS Med* 14, e1002258 (2017).

R1-10 The authors extensively adjusted for confounders, and even performed several sensitivity analyses including self-reported measures of health.

Response: Thanks for your comment. To increase the robustness of our findings, we have conducted four types of sensitivity analyses in addition to the primary analysis (Method section, Page 16, Lines 480-494 in the revised manuscript): “To test the robustness of our results, we performed several sensitivity analyses... Second, to explore the possibility of reverse causation due to impaired cognitive function, which might influence the accuracy of reported lifestyle behaviors, participants with baseline MMSE score below the 10th percentile were excluded³⁵...Fourth, due to common occurrence of missing values for specific items of MMSE test, such as visual construction, the analysis was restricted to participants who complete all the MMSE items⁴².Fifth, as lifestyle score might change during follow-up, latent class trajectory model was used to identify distinct lifestyle score trajectories, and then to assess the effect of lifestyle trajectory groups on cognitive decline. Finally, to account for the non-linear relationship, quadratic terms of time were included in the multivariate model¹⁵.”

R1-11 The authors provide access to their datasets and code.

Response: Thanks for your reminding. The datasets used and analyzed of this study are available from Peking University Open Research Data website (<https://opendata.pku.edu.cn/dataverse/CHADS>). The genetic data underlying this study cannot be shared publicly due to the privacy of individuals that participated in the study, which will be shared on reasonable request to the corresponding author (Prof. Shi: shixm@chinacdc.cn or Associate Prof. Lv: lv Yuebin@nieh.chinacdc.cn). Full annotated SAS code, corresponding datasets for the analysis, and a read-me file can be obtained from <https://github.com/wj2016a/hlsprs.git>. The relevant text has been updated in the Data availability and Code availability parts in the revised manuscript.

R1-12 Surprising to see that there is little change in the visual spatial domain for cognition, this intrigues me since patients at risk for dementia generally miss out points on this domain in an early phase of the disease.

Response: Thanks for your constructive suggestions. As you mentioned, evidence have shown that visual impairment was associated with faster cognitive decline among older adults (*Ref:1-2*), and improving vision status positively impacts the cognitive function (*Ref:3*). One item regarding drawing a picture was used to evaluate the visual construction in this study. Favorable lifestyle was associated with slower decline in visual construction score compared with unfavorable lifestyle, however, the difference did not reach statistical significance (all *P* value >0.05). More items for measuring the visual spatial ability are needed in further research. Additionally, Response R1-14 indicates that missing values occur more frequently among visual construction item than the other items of MMSE test, which might partially explain the null association.

We have emphasized this in the limitations of our study as (*Discussion section, Page 10, Lines 299-303 in the revised manuscript*): “Fifth, given the limited number of items to assess the visual construction and common occurrence of missing values in this domain⁴², nonsignificant association between healthy lifestyle and genetic risk on visual construction should be interpreted with caution. Further studies using multiple indicators to evaluate visuospatial ability are needed.”

Reference:

1. Shang X, Zhu Z, Wang W, Ha J, He M. The Association between Vision Impairment and Incidence of Dementia and Cognitive Impairment: A Systematic Review and Meta-analysis. *Ophthalmology* 128, 1135-1149 (2021).
2. Nagarajan N, et al. Vision impairment and cognitive decline among older adults: a systematic review. *BMJ open* 12, e047929 (2022).
3. Yeo BSY, Ong RYX, Ganasekar P, Tan BKJ, Seow DCC, Tsai ASH. Cataract Surgery and Cognitive Benefits in the Older Person: A Systematic Review and Meta-analysis. *Ophthalmology*, (2024). doi: 10.1016/j.ophtha.2024.02.003

Minor points:

R1-13 Figure 1 is insightful, but also raises questions. Why do all participants start at the same intercept? would it not be more in line with biology to allow for random

intercepts, in order to capture differences at baseline at the advanced age of 81?

Response: Thanks for your constructive suggestion. The intercepts for different groups exhibit a relatively small difference, which may result in an apparent similarity of intercepts in Figure 1. Similar situations were also observed in previous studies (Ref:1-2).

- (1) For Figure 1a, the intercepts were 0.06978, 0.1048, 0.08762, and 0.07997 for high genetic risk & unfavorable lifestyle group, high genetic & favorable lifestyle group, low genetic risk & unfavorable lifestyle group, and low genetic & favorable lifestyle group, respectively (Figure R1-1).
- (2) For Figure 1b, among participants with low genetic risk, the intercepts were 0.00403 for unfavorable lifestyle group, and 0.01507 for favorable lifestyle group, respectively.
- (3) For Figure 1c, among participants with high genetic risk, the intercepts were 0.1437 for unfavorable lifestyle group, and 0.1616 for favorable lifestyle group, respectively.

Figure R1-1. Rate of change in cognitive function according to genetic risk and lifestyle profiles.

Models were adjusted for age, sex, entry time, educational attainment, area of residence, current marital status, occupation, source of income, and baseline cognitive score.

Reference:

1. Levine DA, et al. Associations Between Vascular Risk Factor Levels and Cognitive Decline Among Stroke Survivors. *JAMA network open* 6, e2313879 (2023).
2. Li Y, et al. Pet Ownership, Living Alone, and Cognitive Decline Among Adults 50 Years and Older. *JAMA network open* 6, e2349241 (2023).

R1-14 Note that the authors selected individuals that completed all cognitive tests/visits, this could lead to selection bias. It would be helpful to comment on this matter, and to quantify whether this could have affected results.

Response: Thanks for your advice. Restricted to participants who completed all MMSE items might lead to selection bias and overestimate the association between healthy lifestyle and cognitive decline (Ref:1). Majority of participants were missing only one item, e.g. visual construction, from the MMSE test in this study. Hence, the included participants encompassed both those who completed all MMSE items and those who had partial missing items.

We further conducted sensitivity analysis by restricting to participants who completed all items on MMSE test (N=18159), and the observed effect sizes were consistent with the main analysis (Table R1-7 & Table R1-8).

We have made the following amendments in the revised manuscript:

- (1) Methods section (Methods section, Page 16, Lines 487-489 in the revised manuscript): *“Fourth, due to common occurrence of missing values for specific items of MMSE test, such as visual construction, the analysis was restricted to participants who complete all the MMSE items⁴².”*
- (2) Results section: we have added the results in Supplementary Table S11 & Table S12.
- (3) Discussion section: we have added this in the limitations of our study as (Discussion section, Page 10, Lines 303-308 in the revised manuscript): *“Additionally, restricted to participants who completed all Mini-Mental State Examination (MMSE) items might lead to an overestimation of the association between healthy lifestyle and cognitive decline⁴². However, the results of sensitivity analysis including only participants who completed all MMSE items was consistent with the main analysis.”*

Table R1-7. Sensitivity analyses for association of healthy lifestyle with cognitive decline among overall participants who completed all items of MMSE

Analysis	Unfavorable lifestyle	Intermediate lifestyle		Favorable lifestyle	
		Difference (95% CI)	P value	Difference (95% CI)	P value
Cognitive score	Reference	0.043 (0.021, 0.065)	0.0001	0.158 (0.127, 0.188)	<0.0001
Orientation score	Reference	0.040 (0.013, 0.067)	0.0034	0.176 (0.140, 0.213)	<0.0001
Attention and calculation score	Reference	0.017 (0.006, 0.027)	0.0014	0.048 (0.034, 0.062)	<0.0001
Visual construction score	Reference	0.002 (-0.004, 0.008)	0.5176	0.001 (-0.008, 0.009)	0.8789
Language score	Reference	0.055 (0.034, 0.076)	<0.0001	0.155 (0.126, 0.184)	<0.0001
Naming score	Reference	0.026 (0.013, 0.039)	0.0001	0.068 (0.051, 0.086)	<0.0001
Recall score	Reference	0.013 (0.001, 0.026)	0.0284	0.058 (0.042, 0.074)	<0.0001

Adjusted for age, sex, entry time, educational attainment, area of residence, current marital status, occupation, source of income, and baseline cognitive score. For the analysis of the cognitive score of six dimensions, models were additionally adjusted for the appropriate baseline dimensions of cognitive score instead of baseline cognitive score.

Table R1-8. Sensitivity analyses for association of healthy lifestyle with cognitive decline in different genetic risk groups among participants who completed all items of MMSE

Analysis	Unfavorable Lifestyle	Intermediate Lifestyle		Favorable Lifestyle	
		Difference (95% CI)	P value	Difference (95% CI)	P value
Low genetic risk					
Cognitive score	Reference	0.039 (0.009, 0.069)	0.0117	0.070 (0.032, 0.109)	0.0003
Orientation score	Reference	0.043 (0.008, 0.078)	0.0174	0.074 (0.029, 0.119)	0.0012
Attention and calculation score	Reference	0.017 (0.001, 0.032)	0.0360	0.022 (0.002, 0.042)	0.0288
Visual construction score	Reference	0.005 (-0.007, 0.016)	0.4474	0.005 (-0.010, 0.020)	0.5059
Language score	Reference	0.026 (-0.002, 0.054)	0.0656	0.066 (0.031, 0.101)	0.0002
Naming score	Reference	0.020 (0.002, 0.038)	0.0287	0.022 (-0.001, 0.045)	0.0593

Recall score	Reference	0.019 (0.001, 0.037)	0.0424	0.026 (0.003, 0.049)	0.0268
High genetic risk					
Cognitive score	Reference	0.010 (−0.024, 0.044)	0.5673	0.065 (0.024, 0.107)	0.0022
Orientation score	Reference	0.003 (−0.036, 0.042)	0.8937	0.085 (0.037, 0.133)	0.0005
Attention and calculation score	Reference	0.006 (−0.011, 0.022)	0.5084	0.026 (0.005, 0.046)	0.0141
Visual construction score	Reference	−0.008 (−0.020, 0.003)	0.1652	−0.018 (−0.032, −0.003)	0.0165
Language score	Reference	0.022 (−0.008, 0.053)	0.1502	0.062 (0.024, 0.100)	0.0012
Naming score	Reference	0.007 (−0.012, 0.026)	0.4623	0.023 (−0.000, 0.046)	0.0544
Recall score	Reference	−0.007 (−0.026, 0.012)	0.4840	0.015 (−0.008, 0.039)	0.1928

Adjusted for age, sex, entry time, educational attainment, area of residence, current marital status, occupation, source of income, and baseline cognitive score. For the analysis of the cognitive score of six dimensions, models were additionally adjusted for the appropriate baseline dimensions of cognitive score instead of baseline cognitive score.

Reference:

1. Godin J, Keefe J, Andrew MK. Handling missing Mini-Mental State Examination (MMSE) values: Results from a cross-sectional long-term-care study. *Journal of epidemiology* 27, 163-171 (2017).

R1-15 With this large population it seems to be possible to account for non-linearity, did the authors consider to fit in natural splines, to account for non-linear changes of cognition at this age?

Response: Thanks for your constructive suggestion. To account for the non-linear relationship, quadratic terms of time were included in the multivariate model (*Ref:1*), and results are shown in Figure R1-2. We have also made the following amendments in the revised manuscript:

- (1) Methods section (Methods section, Page 17, Lines 492-494 in the revised manuscript) “Finally, to account for the non-linear relationship, quadratic terms of time were included in the multivariate model¹⁵”.
- (2) Results section (Results section, Page 7, Lines 202-205 in the revised manuscript): “When a quadratic term for time was included in the multivariate linear mixed-effects models, the fitted curves were not perfectly straight, but displayed slightly curved (*Supplementary Fig. 7*).”
- (3) Discussion section: we have added this in the limitations of our study as (Discussion section, Page 11, Lines 320-323 in the revised manuscript): “Eighth, although the possible of non-linear association between lifestyle and cognitive decline has been observed, the fitted curve were highly close to the straight line, which need to be validated further studies.”

Figure R1-2. Predicted non-linear change in cognitive score according to genetic risk and healthy lifestyle

Reference:

1. Dhana K, et al. Genetic risk, adherence to a healthy lifestyle, and cognitive decline in African Americans and European Americans. *Alzheimers Dement* 18, 572-580 (2022).

R1-16 The code is brief, and does not seem to capture all steps in the analysis process and requires a bit more read.me information to follow.

Response: Thanks for your reminding. The code for data analysis has been reorganized, increasing from original 30 lines of SAS code to 1049 lines. Full annotated SAS code, corresponding datasets for data analysis, and a read-me file have been provided and uploaded to GitHub: <https://github.com/wj2016a/hlsprs.git>, including three sections: data clean, model construction, and plot making. First, data about healthy lifestyle score, cognitive score and six dimensions of cognitive score, and covariates (age, sex, educational level, urban, source of income, etc.) has been cleaned in accordance with data cleaning manuals and previous studies. Second, model construction includes the linear mixed model (to explore the effect of healthy lifestyle and genetic risk on subsequent cognitive decline) and Cox proportional hazard regression model (to explore the impact of healthy lifestyle and genetic risk on incident of cognitive impairment). Third, R package “*ggplot2*” was used for making the plot.

Comments of Reviewer 2

R2-1 This study highlights important aspect of cognitive trajectory based on lifestyle and genetic risk subgroups in a longitudinal population-based cohort from the Chinese descent. Study design and methods are extensive and reasonable, while results are convincing. However, the manuscript was missing with some details or misplaced due to the Journal's format, which need to be improved.

Response: Thanks for your constructive comments which helped us to improve the manuscript. Some details have been added to the Methods part and Supplementary materials regarding the information of included SNPs for construction of polygenic risk score, and our effort to fulfill them in the process. Tables and figures are also revised according to the comments. We believe the revision process improves the study to be more convincing and suitable for the journal.

R2-2 Lines 103-104: In the Results section, please include summary of lifestyle (Unfavorable/Intermediate/Favorable) and genetic (low vs high genetic risk) subgroups from Methods since this description came in Methods.

Response: Thanks for your advice. We have added corresponding content to the Results section as follows (Results section, Page 3, Lines 99-106 in the revised manuscript): “Among 18811 participants (mean age 82.97 years, 52.28% women), 10515 (55.90%) were of unfavorable lifestyle, 5885 (31.28%) of intermediate lifestyle, and 2411 (12.82%) of favorable lifestyle. The mean (SD) value of baseline cognitive score was 26.73 (3.24). The present study involved 6301 participants with genetic information, and the polygenic risk score was approximately normal distributed (Figure S2). Participants were divided equally into low genetic risk and high genetic risk groups.”

R2-3 What is the rationale including SNPs associated other diseases?

Response: Thanks for pointing this out. All 34 SNPs included in the polygenic risk score (PRS) was associated with Alzheimer's disease (AD) according to previous genome-wide association studies (GWAS) (Table R2-1, Ref:1-2), including APOE (Ref:3-4).

We have revised our manuscript as follows (Methods section, Page 14, Lines 430-434 in the revised manuscript): “The polygenic risk score in the present study was calculated based on the number of risk alleles of 34 genetic variants increasing the risk for Alzheimer's disease as previously published^{52, 54}. Details regarding the selected SNPs are provided in Supplementary Table S3.”

Table R2-1. List of SNPs included in the polygenic risk score

SNP #	Chromosome	Chromosome position	Gene	Risk Allele	Beta	P value
rs6656401	1	207518704	CR1	A	0.1655	6.00E-24
rs6701713	1	207612944	CR1	A	0.1484	5.00E-10
rs11889338	2	17246318	ZFYVE9P2, RN7SKP168	A	0.4383	9.00E-06
rs7561528	2	127132061	NIFKP9, BIN1	A	0.1570	4.00E-14
rs6733839	2	127135234	NIFKP9, BIN1	T	0.1989	7.00E-44
rs1552244	3	10093893	FANCD2, FANCD2OS	A	0.2776	2.00E-06

rs6448799	4	11628425	HS3ST1, LINC02360	T	0.0770	7.00E-08
rs727153	4	154733269	LRAT	C	0.4886	3.00E-06
rs190982	5	88927603	MEF2C-AS1	A	0.0770	3.00E-08
rs9271192	6	32610753	HLA-DRB1, HLA-DQA1	C	0.1044	3.00E-12
rs9381040	6	41186912	TREM2, TREML2	C	0.0770	6.00E-07
rs6922617	6	41368363	NCR2, FOXP4-AS1	A	0.0900	4.00E-08
rs9349407	6	47485642	CD2AP	C	0.1044	9.00E-09
rs10948363	6	47520026	CD2AP	G	0.0953	5.00E-11
rs1476679	7	100406823	ZCWPW1	T	0.0953	6.00E-10
rs11771145	7	143413669	EPHA1-AS1	G	0.1044	1.00E-13
rs10273775	7	147200311	CNTNAP2	G	0.4187	9.00E-06
rs28834970	8	27337604	PTK2B	C	0.0953	7.00E-14
rs9331896	8	27610169	CLU	T	0.1484	3.00E-25
rs7818382	8	95041772	NDUFAF6	T	0.0677	8.00E-08
rs956225	8	121897448	HAS2-AS1	A	1.2030	9.00E-06
rs514716	9	3929424	GLIS3	G	0.0710	3.00E-09
rs7920721	10	11678309	USP6NL-AS1, ECHDC3	G	0.0677	3.00E-07
rs474951	11	60071148	MS4A3, MS4A2	T	0.2390	1.00E-06
rs10792832	11	86156833	RNU6-560P, LINC02695	G	0.1398	9.00E-26
rs17511627	13	26150190	ATP8A2P3, RNF6	C	0.5596	5.00E-06
rs17125944	14	52933911	FERMT2	C	0.1310	8.00E-09
rs10498633	14	92460608	SLC24A4	G	0.0953	6.00E-09
rs3752246	19	1056493	ABCA7	G	0.1398	6.00E-07
rs4147929	19	1063444	ABCA7	A	0.1398	1.00E-15
rs6859	19	44878777	NECTIN2	A	0.3436	1.00E-07
rs8035452	15	50748601	SPPL2A	T	0.0770	3.00E-07
rs429358*	19	44908684	APOE	T		
rs7412*	19	44908822	APOE	C		

*All the selected SNPs were associated with Alzheimer's disease.

*APOE ϵ 2 and ϵ 4 allele were obtained based on the combination of APOE rs429358 and APOE rs7412 polymorphisms, with the effect value for Alzheimer's disease of -0.47 and 1.03, respectively.

Reference:

1. Liu X, et al. Integrated genetic analyses revealed novel human longevity loci and reduced risks of multiple diseases in a cohort study of 15,651 Chinese individuals. *Aging Cell* 20, e13323 (2021).
2. Desikan RS, et al. Genetic assessment of age-associated Alzheimer disease risk: Development and validation of a polygenic hazard score. *PLoS Med* 14, e1002258 (2017).
3. Li JQ, et al. Risk factors for predicting progression from mild cognitive impairment to Alzheimer's disease: a systematic review and meta-analysis of cohort studies. *J Neurol Neurosurg Psychiatry* 87, 476-484 (2016).
4. Therriault J, et al. Association of Apolipoprotein E epsilon4 With Medial Temporal Tau Independent of Amyloid-beta. *JAMA Neurol* 77, 470-479 (2020).

R2-4 Line 139: Subtitles seemed grammatically wrong. Suggest revising them.

Response: Thanks for pointing this out. We have modified the previous subtitle "Association of lifestyle, genetic risk, with cognitive decline" in the revised manuscript as follows (Results section, Page 4, Lines 121-122; Page 5, Lines 148-149; Page 6, Line 165 in the revised manuscript): "Individual association of healthy lifestyle and genetic risk with cognitive decline; Association of healthy lifestyle with cognitive decline

stratified by genetic risk; and Joint association of healthy lifestyle and genetic risk with cognitive decline ”.

R2-5 Table S6: % difference calculation did not consider point reference’s point estimate. For example, visual construction score difference was 100% between favorable and unfavorable. However, estimate was small (-0.006 vs. -0.001). Interpretation should be cautious.

Response: Thanks for your constructive advice. Adherence to favorable lifestyle tended to slower the decline of visual construction, but this association did not reach the statistically significant level (all P values >0.05). Thus, the results of percentage of visual construction should be considered with caution. It has been listed as an important limitation of our study as (Discussion section, Page 10, Lines 299-303 in the revised manuscript): “Fifth, given the limited number of items to assess the visual construction and common occurrence of missing values in this domain⁴², nonsignificant association between healthy lifestyle and genetic risk on visual construction should be interpreted with caution. Further studies using multiple indicators to evaluate visuospatial ability are needed.”

R2-6 Genetic risk groups: how about just using APOE e4 carrier status. Was PRS based subgrouping better?

Response: Thanks for your constructive suggestions. A higher discrimination power has been achieved by PRS in comparison to APOE $\epsilon 4$ carrier status alone (Figure R2-1).

No significant difference in cognitive decline between participants who carry APOE $\epsilon 4$ allele and in participants who do not (P value=0.56). However, the cognitive decline occurred faster in participants with high genetic risk than those with low genetic risk group (difference, -0.023 SDU per year [95% CI, -0.044 to -0.003 SDU per year], P value =0.0253) when utilizing the PRS. Thus, PRS was used for risk stratification for cognitive decline instead of APOE $\epsilon 4$ carrier status alone in the present study.

Figure R2-1. Rate of change in cognitive score according to genetic risk factors. Models were adjusted for age, sex, entry time, educational attainment, area of residence, current marital status,

occupation, source of income, and baseline cognitive score. a) *APOE ε4* carrier status; b) Polygenic risk score based on 34 SNPs associated with increased risk of Alzheimer’s disease, and divided into two groups according to the median value of polygenic risk score.

R2-7 Figure 2: It is unclear about the joint model including both lifestyle and genetic profiles improved the overall prediction on cognitive decline.

Response: Thanks for your valuable advice. Combining lifestyle and genetic factors produced a better prediction model than using any individual factor. According to the PRS and the lifestyle score, the participants were divided into 6 groups in the analysis of joint effect on cognitive decline. Participants at the high genetic risk with the unfavorable lifestyle served as the reference group (Table R2-2).

Compared with participants with high genetic risk and unfavorable lifestyle, those with low genetic risk and favorable healthy lifestyle experienced the slowest rates of cognitive decline over time. Similar results were observed for most dimensions of cognitive function, except for visual construction. We have added the results to Supplementary information (Table S7).

Table R2-2. Joint effect of genetic risk and lifestyle factors on rate of changes in cognitive decline

	Standard deviation units			Difference, %
	Estimate (95% CI)	Difference (95% CI)	P value	
Cognitive score				
High genetic risk & unfavorable lifestyle	-0.196 (-0.217, -0.175)	Reference		
High genetic risk & intermediate lifestyle	-0.194 (-0.219, -0.169)	0.002 (-0.030, 0.035)	0.8805	1.02
High genetic risk & favorable lifestyle	-0.133 (-0.168, -0.099)	0.063 (0.023, 0.103)	0.0021	32.14
Low genetic risk & unfavorable lifestyle	-0.190 (-0.211, -0.169)	0.007 (-0.023, 0.036)	0.6648	3.57
Low genetic risk & intermediate lifestyle	-0.148 (-0.172, -0.124)	0.048 (0.017, 0.080)	0.0029	24.49
Low genetic risk & favorable lifestyle	-0.108 (-0.144, -0.073)	0.088 (0.047, 0.129)	<0.0001	44.90
Orientation score				
High genetic risk & unfavorable lifestyle	-0.193 (-0.218, -0.169)	Reference		
High genetic risk & intermediate lifestyle	-0.191 (-0.220, -0.162)	0.003 (-0.035, 0.040)	0.8954	1.55
High genetic risk & favorable lifestyle	-0.110 (-0.150, -0.071)	0.083 (0.037, 0.129)	0.0004*	43.01
Low genetic risk & unfavorable lifestyle	-0.181 (-0.206, -0.157)	0.012 (-0.022, 0.046)	0.4812	6.22
Low genetic risk & intermediate lifestyle	-0.137 (-0.165, -0.110)	0.056 (0.019, 0.093)	0.0028*	29.02
Low genetic risk & favorable lifestyle	-0.105 (-0.145, -0.065)	0.088 (0.042, 0.135)	0.0002*	45.60
Attention and calculation score				
High genetic risk & unfavorable lifestyle	-0.080 (-0.091, -0.070)	Reference		
High genetic risk & intermediate lifestyle	-0.076 (-0.088, -0.063)	0.005 (-0.012, 0.021)	0.5836	6.25
High genetic risk & favorable lifestyle	-0.056 (-0.073, -0.039)	0.024 (0.004, 0.044)	0.0188	30.00
Low genetic risk & unfavorable lifestyle	-0.076 (-0.087, -0.066)	0.004 (-0.011, 0.019)	0.6026	5.00
Low genetic risk & intermediate lifestyle	-0.059 (-0.071, -0.047)	0.021 (0.006, 0.037)	0.0082	26.25
Low genetic risk & favorable lifestyle	-0.052 (-0.070, -0.035)	0.028 (0.008, 0.048)	0.0072*	35.00
Visual construction score				
High genetic risk & unfavorable lifestyle	0.003 (-0.005, 0.011)	Reference		
High genetic risk & intermediate lifestyle	-0.005 (-0.014, 0.004)	-0.008 (-0.020, 0.004)	0.1725	NA

High genetic risk & favorable lifestyle	-0.015 (-0.027, -0.002)	-0.018 (-0.032, -0.003)	0.0168	
Low genetic risk & unfavorable lifestyle	-0.003 (-0.010, 0.005)	-0.006 (-0.017, 0.005)	0.3064	
Low genetic risk & intermediate lifestyle	0.002 (-0.007, 0.011)	-0.001 (-0.013, 0.011)	0.8801	
Low genetic risk & favorable lifestyle	0.003 (-0.010, 0.015)	-0.001 (-0.015, 0.014)	0.9467	
Language score				
High genetic risk & unfavorable lifestyle	-0.155 (-0.174, -0.136)	Reference		
High genetic risk & intermediate lifestyle	-0.133 (-0.156, -0.111)	0.021 (-0.008, 0.051)	0.1567	13.55
High genetic risk & favorable lifestyle	-0.095 (-0.126, -0.064)	0.060 (0.024, 0.096)	0.0012*	38.71
Low genetic risk & unfavorable lifestyle	-0.135 (-0.154, -0.116)	0.020 (-0.007, 0.047)	0.1443	12.90
Low genetic risk & intermediate lifestyle	-0.109 (-0.130, -0.087)	0.046 (0.017, 0.075)	0.0016*	29.68
Low genetic risk & favorable lifestyle	-0.067 (-0.098, -0.035)	0.088 (0.051, 0.125)	<0.0001*	56.77
Naming score				
High genetic risk & unfavorable lifestyle	-0.065 (-0.078, -0.053)	Reference		
High genetic risk & intermediate lifestyle	-0.058 (-0.073, -0.044)	0.007 (-0.011, 0.026)	0.4538	10.77
High genetic risk & favorable lifestyle	-0.042 (-0.062, -0.023)	0.023 (0.000, 0.046)	0.0479	35.38
Low genetic risk & unfavorable lifestyle	-0.059 (-0.071, -0.047)	0.007 (-0.011, 0.024)	0.4550	10.77
Low genetic risk & intermediate lifestyle	-0.039 (-0.052, -0.025)	0.027 (0.009, 0.045)	0.0039*	41.54
Low genetic risk & favorable lifestyle	-0.037 (-0.057, -0.017)	0.029 (0.005, 0.052)	0.0160	44.62
Recall score				
High genetic risk & unfavorable lifestyle	-0.078 (-0.090, -0.066)	Reference		
High genetic risk & intermediate lifestyle	-0.086 (-0.100, -0.071)	-0.008 (-0.026, 0.011)	0.4251	NA
High genetic risk & favorable lifestyle	-0.063 (-0.083, -0.043)	0.015 (-0.008, 0.038)	0.1971	19.23
Low genetic risk & unfavorable lifestyle	-0.078 (-0.090, -0.065)	0.000 (-0.017, 0.018)	0.9566	0
Low genetic risk & intermediate lifestyle	-0.059 (-0.073, -0.045)	0.019 (0.001, 0.037)	0.0420	24.36
Low genetic risk & favorable lifestyle	-0.052 (-0.072, -0.031)	0.026 (0.003, 0.050)	0.0273	33.33

Models were adjusted for age, sex, entry time, educational attainment, area of residence, current marital status, occupation, source of income, and baseline cognitive score. For the analysis of six cognitive dimensions, models were additionally adjusted for the baseline dimensions of cognitive score as appropriate instead of baseline cognitive score.

* Indicating statistically significant Bonferroni corrected P value ($P < 0.008$).

R2-8 Line 163: This analysis seemed related to progression to cognitive impairment. Subheading was misleading as a cross-sectional analysis. Suggest changing it.

Response: Thank you for pointing this out. We have modified the subtitle in the revised manuscript as follows (Results section, Page 6, Lines 182-183 in the revised manuscript): “Association of healthy lifestyle and genetic risk with risk of incident cognitive impairment”.

R2-9 Lines 178-181: Sensitivity analysis tables were very difficult to follow. Any significant results? Please summarize key findings with p values.

Response: Thanks for the reminding. We have added the corresponding P values for each estimate in sensitivity analyses (Supplementary Table S11-S13), and statistically significant results were highlighted with asterisks, making the results will be much easier to follow. Similar estimates were obtained, which confirmed the findings of the main analysis.

R2-10 Lines 180-192: “Healthy lifestyle might outweigh genetic background in relation to cognitive decline”. This statement may not be fair. Since, Also, genetic risk subgroups are dependent on the list of risk SNPs and distribution of the PRS. This study used SNPs associated with other diseases and used median of PRS distribution for subgrouping. If you choose AD associated SNPs and more extremes of the PRS distribution using the AD associated, you may see larger % differences between genetic subgroups.

Response: Thanks for your insightful comments. Individuals were classified into low and high genetic risk categories based on a set of percentiles of the PRS to investigate the stratification of the PRS, using the following five classifications (Figure R2-2, Ref:1-2): (1) low (bottom 50%) and high (top 50%); (2) low (bottom 60%) and high (top 40%); (3) low (bottom 70%) and high (top 30%); (4) low (bottom 80%) and high (top 20%); (5) low (bottom 90%) and high (top 10%).

As the genetic risk groups were identified by the extremes of PRS distribution, the rate of change in cognitive decline was significantly greater %differences between genetic subgroups. There was an approximately two-fold increase in estimated difference of change in cognitive function between low and high genetic risk groups using 10th percentile cut points of PRS than using the median value of PRS (difference, 0.048 SDU [95%CI, 0.014–0.082 SDU per year], $P=0.0052$ vs difference, 0.023 SDU [95%CI, 0.003–0.044 SDU per year], $P=0.0253$). However, the effect size of genetic risk factors on cognitive decline was relatively smaller compared to the favorable lifestyle (difference, 0.176 SDU [95%CI, 0.147–0.205 SDU per year], $P<0.0001$). Given the sample size in each group in analysis of joint effect of genetic risk and lifestyle factors (six groups), participants were divided into two genetic risk groups according to the median value of PRS to guarantee sufficient statistical power (Ref:3-5).

Additionally, we have listed this as an important limitation in the revised manuscript (Discussion section, Page 10, Lines 308-315 in the revised manuscript): “*Sixth, our genetic risk score did not include all the variants that associated with cognitive decline, which might attenuate the true effect. Future studies should construct genetic risk score of cognitive decline by comprehensive coverage of established cognitive decline related genetic variants to improve the accuracy of risk classification. Additional studies with larger sample size are warranted to verify our findings and examine the impact of genetic risk factors on cognitive decline at the extreme end of the polygenic risk score distribution.*”

Figure R2-2. Rate of change in cognitive score according to different genetic risk categories. Models were adjusted for age, sex, entry time, educational attainment, area of residence, current marital status, occupation, source of income, and baseline cognitive score. a) PRS categories were defined as low (bottom 50%), and high (top 50%) by the 50th percentiles of the PRS; b) PRS categories were defined as low (bottom 60%), and high (top 40%) by the 40th percentiles of the PRS; c) PRS categories were defined as low (bottom 70%), and high (top 30%) by the 30th percentiles of the PRS; d) PRS categories were defined as low (bottom 80%), and high (top 20%) by the 20th percentiles of the PRS; e) PRS categories were defined as low (bottom 90%), and high (top 10%) by the 10th percentiles of the PRS.

Reference:

1. Lu X, et al. A polygenic risk score improves risk stratification of coronary artery disease: a large-scale prospective Chinese cohort study. *Eur Heart J* 43, 1702-1711 (2022).
2. Cui Q, et al. Integrating polygenic and clinical risks to improve stroke risk stratification in prospective Chinese cohorts. *Sci China Life Sci* 66, 1626-1635 (2023).
3. Wang J, et al. Healthy lifestyle in late-life, longevity genes, and life expectancy among older adults: a 20-year, population-based, prospective cohort study. *Lancet Healthy Longev* 4, e535-e543 (2023).
4. Wang L, Xie J, Hu Y, Tian Y. Air pollution and risk of chronic obstructed pulmonary disease: The modifying effect of genetic susceptibility and lifestyle. *EBioMedicine* 79, 103994 (2022).
5. Hayes JP, et al. Genetic Risk for Alzheimer Disease and Plasma Tau Are Associated With Accelerated Parietal Cortex Thickness Change in Middle-Aged Adults. *Neurol Genet* 9, e200053 (2023).

R2-11 APOE e4 vs. PRS: the current study did not show whether subgrouping with PRS approach is better than that with the APOE e4 allele alone.

Response: Thanks for your advice. A higher discrimination power for cognitive decline has been achieved by PRS in comparison to *APOE* $\epsilon 4$ carrier status alone. Please refer to previous response **R2-6** for more specific details.

Comments of Reviewer 3

R3-1 I think this is an important study, re-confirming the role of lifestyle on cognition and showing that a healthy lifestyle may attenuate the genetic risk. Overall, it is a well-conducted study, and I have only a few comments.

Response: Thanks for your positive and constructive comments on our manuscript. We have given point-by-point responses to your comments as below and revised the manuscript according to your constructive suggestions. All indicated page numbers in this document refer to the manuscript file with the highlighted changes.

R3-2 Does the lifestyle change during the follow-up? I suggest showing a figure for trajectories of lifestyle scores during the follow-up.

Response: Thanks for your constructive suggestions. Changes in the lifestyle score over time was evaluated by latent class trajectory model, and relatively minor changes were observed over time, indicating longitudinal lifestyle score remaining relatively stable during the follow-up.

Three distinct trajectories of lifestyle score were identified (Table R3-1, Figure R3-1), named as low-stable trajectory, intermediate-stable trajectory, and high-stable trajectory. We further evaluate the impact of lifestyle trajectory groups on cognitive decline in sensitivity analysis (Ref:1). Participants with stable intermediate and stable high lifestyle score had slower cognitive decline compared with participants with stable low lifestyle score, which followed similar patterns with the main analysis (Table R3-2).

We have modified our manuscript as follows:

- (1) Methods section (Methods section, Page 16, Lines 489-492 in the revised manuscript): *“Fifth, as lifestyle score might change during follow-up, latent class trajectory model was used to identify distinct lifestyle score trajectories, and then to assess the effect of lifestyle trajectory groups on cognitive decline.”*
- (2) Results section: we have added the figure of trajectories of lifestyle score during follow-up as Figure S6, and the results of association between trajectories of lifestyle score and rate of cognitive decline as Table S13 in Supplementary information.
- (3) Discussion section: we have listed this as a key limitation in the revised manuscript (Discussion section, Page 10, Lines 292-296 in the revised manuscript): *“Third, participants were categorized according to baseline lifestyle score, which could not capture the long-term cumulative effects of lifestyle factors and might underestimate the association⁴¹. However, the impact of lifestyle trajectory groups on cognitive decline was evaluated, and results were in line with the main analysis.”*

Table R3-1. The decision for number of lifestyle score trajectory groups

	Number of classes*			
	1	2	3	4
The maximum Log-Likelihood	-72290.80	-72290.80	-72063.11	-72266.03
The Bayesian information criterion	144680.00	144719.40	144303.40	144748.60
Proportion of participants per class				
%Class1	100.00	56.02	14.20	49.34
%Class2	NA	43.98	58.78	23.73

%Class3	NA	NA	27.02	4.94
%Class4	NA	NA	NA	21.99
Mean posterior probabilities for each class				
Mean prob_class 1	1.00	0.51	0.86	0.36
Mean prob_class 2	NA	0.51	0.82	0.33
Mean prob_class 3	NA	NA	0.74	0.28
Mean prob_class 4	NA	NA	NA	0.29

*The latent class trajectory model was fitted using the R package “*lcmm*” to identify trajectories of lifestyle score over time. Non-linearity in the long-term lifestyle score trajectories was tested by adding a quadratic term for follow-up years into the model. The best-fitting number of trajectories were determined by the following criteria: (1) the optimal number of classes based on the lowest Bayesian information criterion (BIC) was chosen; (2) the average of posterior probability of assignment should be greater than 0.7 for all classes, which indicate adequate internal reliability; (3) there was no less than 2% of the population in each trajectory group.

The best fitting model is highlighted in bold characters.

NA: not applicable.

Figure R3-1. Trajectories of healthy lifestyle score for older adults during follow-up. The solid lines show the estimated values of healthy lifestyle score for members in the groups. Shading around the lines represent confidence bands for the calculated trajectory.

Table R3-2. Estimated change in cognitive function by trajectories of lifestyle score

Trajectories of lifestyle score	Standard deviation units			Difference, %
	Estimate (95% CI)	Difference (95% CI)	P value	
Cognitive score				
Low-stable	-0.402 (-0.428, -0.376)	Reference		Reference
Intermediate-stable	-0.368 (-0.381, -0.355)	0.034 (0.005, 0.063)	0.0202	8.46
High-stable	-0.223 (-0.242, -0.205)	0.179 (0.147, 0.210)	<0.0001	44.53
Orientation score				
Low-stable	-0.402 (-0.433, -0.370)	Reference		Reference
Intermediate-stable	-0.368 (-0.384, -0.352)	0.033 (-0.002, 0.069)	0.0637	8.21
High-stable	-0.221 (-0.244, -0.199)	0.180 (0.142, 0.219)	<0.0001	44.78
Attention and calculation score				
Low-stable	-0.145 (-0.157, -0.133)	Reference		Reference
Intermediate-stable	-0.131 (-0.137, -0.125)	0.015 (0.001, 0.028)	0.0341	10.34

High-stable	-0.081 (-0.089, -0.072)	0.064 (0.050, 0.079)	<0.0001	44.14
Visual construction score				
Low-stable	-0.014 (-0.021, -0.006)	Reference		Reference
Intermediate-stable	-0.015 (-0.019, -0.012)	-0.002 (-0.010, 0.007)	0.6747	NA
High-stable	-0.004 (-0.009, 0.001)	0.010 (0.000, 0.019)	0.0399	71.43
Language score				
Low-stable	-0.349 (-0.374, -0.324)	Reference		Reference
Intermediate-stable	-0.305 (-0.317, -0.292)	0.044 (0.016, 0.072)	0.0018	12.61
High-stable	-0.174 (-0.192, -0.157)	0.174 (0.144, 0.205)	<0.0001	49.86
Naming score				
Low-stable	-0.166 (-0.182, -0.151)	Reference		Reference
Intermediate-stable	-0.146 (-0.153, -0.138)	0.020 (0.003, 0.038)	0.0204	12.05
High-stable	-0.082 (-0.092, -0.071)	0.084 (0.066, 0.103)	<0.0001	50.60
Recall score				
Low-stable	-0.165 (-0.180, -0.151)	Reference		Reference
Intermediate-stable	-0.149 (-0.156, -0.142)	0.016 (0.000, 0.032)	0.0444	9.70
High-stable	-0.088 (-0.098, -0.078)	0.078 (0.060, 0.095)	<0.0001	47.27

Models were adjusted for age, sex, entry time, educational attainment, area of residence, current marital status, occupation, source of income, and baseline cognitive score. For the analysis of the cognitive score of six dimensions, models were additionally adjusted for the appropriate baseline dimensions of cognitive score instead of baseline total cognitive score.

Reference:

1. Sakaniwa R, et al. Trajectories of renal biomarkers and new-onset heart failure in the general population: Findings from the PREVEND study. *Eur J Heart Fail* 25, 1072-1079 (2023).

R3-3 MMSE is limited in determining the cognitive impairment or evaluating the change in those with poor cognition. Maybe sensitivity analysis in which authors exclude the bottom 10% of people with low scores?

Response: Thanks for your advice. We further conducted a sensitivity analysis by excluding the bottom 10% of participants with low MMSE score (Table R3-3 & R3-4), and the results revealed that the impact of healthy lifestyle and genetic risk on cognitive decline were almost unchanged.

We have modified the manuscript as follows:

- (1) Methods section (Method section, Page 16, Lines 480-483 in the revised manuscript): “Second, to explore the possibility of reverse causation due to impaired cognitive function, which might influence the accuracy of reported lifestyle behaviors, participants with baseline MMSE score below the 10th percentile were excluded³⁵.”
- (2) Results section: we have added the corresponding results in Table S11 & Table S12 in Supplementary information.

Table R3-3. Sensitivity analyses for association of healthy lifestyle with cognitive decline among overall participants by excluding participants with baseline MMSE in the lowest 10% of the cohort distribution

Analysis	Unfavorable lifestyle	Intermediate lifestyle		Favorable lifestyle	
		Difference (95% CI)	P value	Difference (95% CI)	P value
Cognitive score	Reference	0.067 (0.046, 0.089)	<0.0001	0.177 (0.148, 0.206)	<0.0001
Orientation score	Reference	0.062 (0.036, 0.087)	<0.0001	0.178 (0.143, 0.212)	<0.0001
Attention and calculation score	Reference	0.027 (0.017, 0.037)	<0.0001	0.060 (0.047, 0.073)	<0.0001
Visual construction score	Reference	0.004 (−0.003, 0.010)	0.2708	0.004 (−0.004, 0.012)	0.3140
Language score	Reference	0.077 (0.057, 0.098)	<0.0001	0.161 (0.134, 0.188)	<0.0001
Naming score	Reference	0.033 (0.020, 0.045)	<0.0001	0.074 (0.057, 0.091)	<0.0001
Recall score	Reference	0.028 (0.016, 0.040)	<0.0001	0.069 (0.053, 0.084)	<0.0001

Models were adjusted for age, sex, entry time, educational attainment, area of residence, current marital status, occupation, source of income, and baseline cognitive score. For the analysis of the cognitive score of six dimensions, models were additionally adjusted for the appropriate baseline dimensions of cognitive score instead of baseline cognitive score.

Table R3-4. Sensitivity analyses for association of healthy lifestyle with cognitive decline in different genetic risk groups by excluding participants with baseline MMSE in the lowest 10% of the cohort distribution

Analysis	Unfavorable Lifestyle	Intermediate Lifestyle		Favorable Lifestyle	
		Difference (95% CI)	P value	Difference (95% CI)	P value
Low genetic risk					
Cognitive score	Reference	0.047 (0.016, 0.078)	0.0030	0.087 (0.048, 0.126)	<0.0001
Orientation score	Reference	0.048 (0.013, 0.082)	0.0068	0.077 (0.034, 0.121)	0.0005
Attention and calculation score	Reference	0.020 (0.004, 0.035)	0.0137	0.026 (0.007, 0.046)	0.0091
Visual construction score	Reference	0.006 (−0.006, 0.018)	0.3565	0.007 (−0.008, 0.022)	0.3691
Language score	Reference	0.033 (0.005, 0.061)	0.0190	0.072 (0.037, 0.107)	0.0001
Naming score	Reference	0.020 (0.002, 0.038)	0.0296	0.026 (0.003, 0.049)	0.0297
Recall score	Reference	0.022 (0.004, 0.040)	0.0171	0.032 (0.009, 0.055)	0.0061
High genetic risk					
Cognitive score	Reference	0.004 (−0.030, 0.037)	0.8328	0.066 (0.024, 0.107)	0.0018
Orientation score	Reference	0.002 (−0.036, 0.040)	0.9177	0.076 (0.030, 0.122)	0.0013
Attention and calculation score	Reference	0.006 (−0.010, 0.023)	0.4611	0.028 (0.008, 0.049)	0.0064
Visual construction score	Reference	−0.008 (−0.020, 0.004)	0.1927	−0.018 (−0.033, −0.003)	0.0173
Language score	Reference	0.022 (−0.008, 0.052)	0.1530	0.061 (0.025, 0.097)	0.0010
Naming score	Reference	0.010 (−0.009, 0.029)	0.3098	0.027 (0.004, 0.050)	0.0204
Recall score	Reference	−0.006 (−0.025, 0.013)	0.5136	0.017 (−0.006, 0.040)	0.1538

Models were adjusted for age, sex, entry time, educational attainment, area of residence, current marital status, occupation, source of income, and baseline cognitive score. For the analysis of the cognitive score of six dimensions, models were additionally adjusted for the appropriate baseline dimensions of cognitive score instead of baseline cognitive score.

R3-4 When investigating each domain, e.g., visual, language, etc., it may require p-value adjustment for multiple testing.

Response: Thank you for pointing this out. We have clarified this in the revised manuscript (Method section, Page 17, Lines 494-497 in the revised manuscript): “Two-sided $P < 0.05$ was accepted as statistically significant, except separate analysis for

individual domains of cognition in which the Bonferroni correction was applied to account for multiple testing ($P < 0.0083$ considered significant [$= 0.05/6$]).”

R3-5 The genetic risk seems to have a negligible effect on cognition, only 0.007 SDU. A comment in a discussion comparing to other studies in the literature is necessary.

Response: Thank you for constructive suggestions. We found that the effect of low genetic risk alone on cognitive decline was smaller compared with a healthier lifestyle. The possible reasons that genetic factors only explained part of the phenotypic variation of cognitive decline could be as follows:

- (1) Prior studies have reported epigenetic changes, such as DNA methylation, might altered the transcription of risk genes for Alzheimer’s disease (*Ref:1-2*).
- (2) Our polygenic risk score did not include all the variants that associated with cognitive decline, which might attenuate the true effect. Further studies that construct genetic risk score of cognitive decline by comprehensive coverage of established cognitive decline related genetic variants to improve the accuracy of risk classification (*Ref:1*).

We have made corresponding amendment in the Discussion section of the revised manuscript as follows(Discussion section, Page 9, Lines 258-260; Page 10, Lines 308-315 in the revised manuscript): *“The modest effects of genetic factors on cognitive decline may be due in part to epigenetic changes that altered the transcription of risk genes for Alzheimer’s disease^{14, 34} ...Sixth, our genetic risk score did not include all the variants that associated with cognitive decline, which might attenuate the true effect. Future studies should construct genetic risk score of cognitive decline by comprehensive coverage of established cognitive decline related genetic variants to improve the accuracy of risk classification. Additional studies with larger sample size are warranted to verify our findings and examine the impact of genetic risk factors on cognitive decline at the extreme end of the polygenic risk score distribution.”*

Reference:

1. Pan G, et al. The potential roles of genetic factors in predicting ageing-related cognitive change and Alzheimer's disease. *Ageing research reviews* 70, 101402 (2021).
2. Rabaneda-Bueno R, Mena-Montes B, Torres-Castro S, Torres-Carrillo N, Torres-Carrillo NM. *Advances in Genetics and Epigenetic Alterations in Alzheimer's Disease: A Notion for Therapeutic Treatment. Genes (Basel)* 12, 1959 (2021).

Responses to reviewers' comments

Reviewer #1 (Remarks to the Author):

Thank you for your extensive and clear comments. These have definitively contributed to improve the quality of the paper. I would strongly advice to use your non-linear version of Figure R1-2 in the main paper, while including confidence intervals. I particularly value your efforts to include a systematic review to stress the novelty of their findings. Thank you for the opportunity to review this paper.

Response: Thanks for your constructive suggestions. We have moved the original Figure R1-2 to the main Figure 3 as suggested in the revised manuscript.

Fig. 3 | Predicted non-linear change in cognitive function according to genetic risk and lifestyle profiles. Linear mixed-effects models were used with adjustment for age, sex, entry time, educational attainment, area of residence, current marital status, occupation, source of income, and baseline cognitive score. Potential non-linear relationship was estimated by adding quadratic terms of time to the multivariate model. a) among participants with low genetic, the non-linear association of lifestyle with cognitive decline; b) among participants with high genetic, the non-linear association of lifestyle with cognitive decline. Solid red line represents point estimates of unfavorable lifestyle group, dashed blue line represents point estimates of favorable lifestyle group, and shaded areas show 95% CIs.